# Learning a Transferable Scheduling Policy for Various Vehicle Routing Problems based on Graph-centric Representation Learning

## Abstract

Reinforcement learning has been used to learn to solve various routing problems. however, most of the algorithm is restricted to finding an optimal routing strategy for only a single vehicle. In addition, the trained policy under a specific target routing problem is not able to solve different types of routing problems with different objectives and constraints. This paper proposes an reinforcement learning approach to solve the min-max capacitated multi vehicle routing problem (mCVRP), the problem seeks to minimize the total completion time for multiple vehicles whose one-time traveling distance is constrained by their fuel levels to serve the geographically distributed customer nodes. The method represents the relationships among vehicles, customers, and fuel stations using relationship-specific graphs to consider their topological relationships and employ graph neural network (GNN) to extract the graph's embedding to be used to make a routing action. We train the proposed model using the random mCVRP instance with different numbers of vehicles, customers, and refueling stations. We then validate that the trained policy solve not only new mCVRP problems with different complexity (weak transferability but also different routing problems (CVRP, mTSP, TSP) with different objectives and constraints (storing transferability).

## 1 Introduction

The Vehicle Routing Problem (VRP), a well-known NP-hard problem, has been enormously studied since it appeared by Dantzig & Ramser (1959). There have been numerous attempts to compute the exact (optimal) or approximate solutions for various types of vehicle routing problems by using mixed integer linear programming (MILP), which uses mostly a branch-and-price algorithm appeared in Desrochers et al. (1992) or a column generation method (Chabrier, 2006), or heuristics ((Cordeau et al., 2002; Clarke & Wright, 1964; Gillett & Miller, 1974; Gendreau et al., 1994)). However, these approaches typically require huge computational time to find the near optimum solution. For more information for VRP, see good survey papers (Cordeau et al., 2002; Toth & Vigo, 2002).

There have been attempts to solve such vehicle routing problems using learning based approaches. These approaches can be categorized into supervised-learning based approaches and reinforcement-learning based approaches (Bengio et al., 2020); supervised learning approaches try to map a target VRP with a solution or try to solve sub-problems appears during optimization procedure, while reinforcement learning (RL) approaches seek to learn to solve routing problems without supervision (i.e, solution) but using only repeated trials and the associated reward signal. Furthermore, the RL approaches can be further categorized into improvement heuristics and construction heuristics (Mazyavkina et al., 2020); improvement heuristics learn to modify the current solution for a better solution, while construction heuristics learn to construct a solution in a sequential decision making framework. The current study focuses on the RL-based construction heuristic for solving various routing problems.

Various RL-based solution construction approaches have been employed to solve the traveling salesman problem (TSP) (Bello et al., 2016; Khalil et al., 2017; Nazari et al., 2018; Kool et al., 2018) or the capacitated vehicle routing problem (CVRP) (Nazari et al., 2018; Kool et al., 2018). (Bello et al., 2016; Nazari et al., 2018; Kool et al., 2018) has used the encoder-decoder structure to

sequentially generate routing schedules, and (Khalil et al., 2017) uses graph based embedding to determine the next assignment action. Although these approaches have shown the potential that the RL based approaches can learn to solve some types of routing problems, these approaches have the major two limitations: (1) only focus on routing a single vehicle over cities for minimizing the total traveling distance (i.e., min-sum problem) and (2) the trained policy for a specific routing problem cannot be used for solving other routing problems with different objective and constraints (they show that trained policy can be used to solve the same type of the routing problems with different problem sizes).

In this study, We proposed the Graph-centric RL-based Transferable Scheduler (GRLTS) for various vehicle routing problems. GRLTS is composed of graph-centric representation learning and RL-based scheduling policy learning. GRLTS is mainly designed to solve min-max capacititated multi vehicle routing problems (mCVRP); the problem seeks to minimize the total completion time for multiple vehicles whose one-time traveling distance is constrained by their fuel levels to serve the geographically distributed customer nodes. The method represents the relationships among vehicles, customers, and fuel stations using relationship-specific graphs to consider their topological relationships and employ graph neural network (GNN) to extract the graph's embedding to be used to make a routing action. To effectively train the policy for minimizing the total completion time while satisfying the fuel constraints, we use the specially designed reward signal in RL framework. The representation learning for graph and the decision making policy are trained in an end-to-end fashion in an MARL framework. In addition, to effectively explore the joint combinatorial action space, we employ curriculum learning while controlling the difficulty (complexity) of a target problem.

The proposed GRLTS resolves the two issues raised in other RL-based routing algorithms:

- GRLTS learns to coordinate multiple vehicles to minimize the total completion time (makespan). It can resolve the first issue of other RL-based routing algorithms and can be used to solve practical routing problems of scheduling multiple vehicles simultaneously. (Kang et al., 2019) also employed the graph based embedding (random graph embedding) to solve identical parallel machine scheduling problem, the problem seeking to minimize the makespan by scheduling multiple machines. However, our approach is more general in that it can consider capacity constraint and more fast and scalable node embedding strategies.

- GRLTS transfers the trained scheduling policy with random mCVRP instances to be used for solving not only new mCVRP problems with different complexity but also different routing problems (CVRP, mTSP, TSP) with different objectives and constraints.

## 2  FORMULATION

### 2.1  MIN-MAX SOLUTION FOR MCVRP

We define the set of vehicles $V_V = 1, ..., N_V$, the set of customers $V_C = 1, ..., N_C$, and the set of refueling stations $V_R = 1, ..., N_R$, where $N_A$, $N_C$, and $N_R$ are the numbers of vehicles, customers, and refueling stations, respectively. The objective of min-max mCVRP is minimizing the makespan that is the longest distance among all vehicle's traveling distance, i.e., $\min \max_{i \in V_V} L_i$ with $L_i$ being the traveling distance of vehicle $i$, while each vehicle's one-time traveling distance is constrained by its remaining fuel. The detailed mathematical formulation using mixed integer linear programming (MILP) is provided in Appendix. Figure (1) (left) shows a snapshot of a mCVRP state and Figure (1) (right) represents a feasible solution of the mCVRP.

### 2.2  DEC-MDP FORMULATION FOR MCVRP

We seek to sequentially construct an optimum solution. Thus, we frame the solution construction procedure as a decentralized Markov decision problem (Dec-MDP) as follows.

#### 2.2.1  STATE

We define the vehicle state $s_t^v, \forall v \in V_V$, the customer state $s_t^c, \forall c \in V_C$, and the refueling station state $s_t^r, \forall r \in V_R$ as follows:

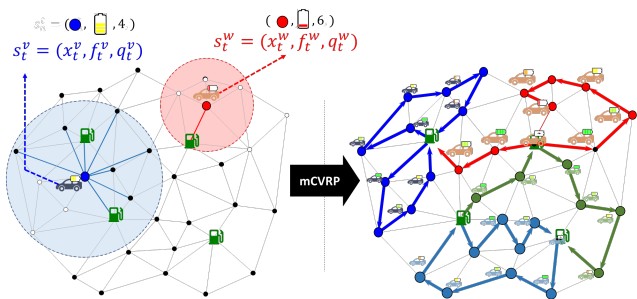

Figure 1: An example of mCVRP. (Left) a snapshot of environment state at time $t$. The circular range of each vehicle indicates the possible moving range with the current fuel level. (Right) a feasible solution of the environment example.

- State of vehicle $v$, $s_t^v = (x_t^v, f_t^v, q_t^v)$. $x_t^v$ is the allocated node that vehicle $v$ to visit; $f_t^v$ is the current fuel level; and $q_t^v$ is the number of customers served by the vehicle $v$ so far.
- State of a customer $c$, $s_t^c = (x^c, \mathbb{v}^c)$. $x^c$ is the location of customer node $c$ (static). Visit indicator $\mathbb{v}^c \in \{0, 1\}$ becomes 1 if the customer $c$ is visited and 0, otherwise.
- State of a refueling station $r$, $s_t^r = x^r$. $x^r$ is the location of the refueling station $r$ (static).

The global state $\mathbf{s}_t$ then becomes $\mathbf{s}_t = (\{s_t^v\}_{v=1}^{N_v}, \{s_t^c\}_{c=1}^{N_C}, \{s_t^r\}_{r=1}^{N_R})$.

### 2.2.2 Actions & State transition

Action $a_t^v$ for vehicle $v$ at time $t$ is indicating a node to be visited by vehicle $v$ at time $t + 1$, that is, $a_t^v = x_{t+1}^v \in \{V_C \cup V_R\}$. Therefore, the next state of vehicle $v$ becomes $s_{t+1}^v = (x_{t+1}^v, f_{t+1}^v, q_{t+1}^v)$ where $f_{t+1}^v$ and $q_{t+1}^v$ are determined deterministically by an action $a_t^v$ as follows:

- Fuel capacity update: $f_{t+1}^v = \begin{cases} F^v, & \text{if } a_t^v \in V_R \\ f_{tv} - d(x_t^v, a_t^v), & \text{otherwise.} \end{cases}$

- Customer visit number update: $q_{t+1}^v = \begin{cases} q_t^v, & \text{if } a_t^v \in V_R \\ q_t^v + 1, & \text{otherwise.} \end{cases}$

### 2.2.3 Rewards

The goal of mCVRP is to force all agents to coordinate to finish the distributed tasks quickly while satisfying the fuel constraints. To achieve this global goal in a distributed manner, we use the specially designed independent reward for each agent as:

- **visiting reward**: To encourage vehicles to visit the customer nodes faster, in turn, minimizing makespan, we define customer visit reward $r_{visit}^v = q_t^v$. This reward is provided when an agent visits a customer; the more customer nodes a vehicle agent $n$ visits, the greater reward it can earn.
- **Refueling reward**: To induce a strategic refueling, we introduce refuel reward $r_{refuel}^v = q_t^v \times ((F^v - f_t^v)/(F^v - 1))^\alpha$. We define the refuel reward as an opportunity cost. That is, vehicles with sufficient fuel are not necessary to refuel (small reward). In contrast, refueling vehicles with a lack of fuel is worth as much as visiting customers. In this study, we set $F^v = 10$ (which is the equivalent to the total traveling distance that vehicle $v$ can travel with the fuel tank fully loaded) and $\alpha = 2$.

## 2.3 Relationships with other class of VRPs

mCVRP, the target problem of this study, has three key properties: 1) the problem seeks to minimize the total completion time of vehicles by forcing all vehicles to coordinate (in a distributed manner), 2) the problem employs fuel capacity constraints requiring the vehicles to visit the refueling stations strategically, 3) the problem considers multiple refueling depots (revisit allowed).

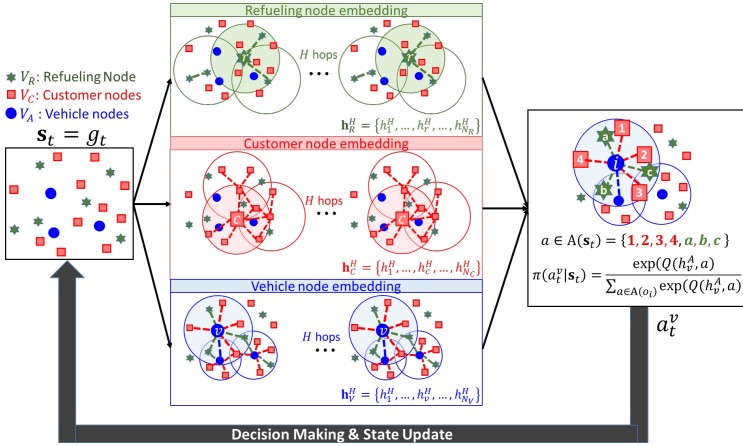

Figure 2: Sequential decision-making framework with trained GRLTS.

If some of these requirements are relaxed, min-max mCVRP can be degenerated into simpler conventional routing problems:

- TSP is the problem where a single vehicle is operated to serve every customer while minimizing the total traveling distance. The agent needs or needs not come back to the depot. This problem does not have capacity constraints.

- CVRP (capacity-constrained TSP) is the problem where a single vehicle is operated to serve every customer while minimizing the total traveling distance and satisfying the fuel constraint. The vehicles need to comeback depot to charge.

- mTSP (multi-agent TSP) is the problem where multiple vehicles should serve all the customers as quickly as possible. This problem does not have capacity constraints.

- mCVRP (multi-agent, capacity-constrained TSP) is our target problem having the properties of both mTSP and CVRP. Additionally, we add more than one refueling depot.

The mathematical formulations for these problems are provided in Appendix. We train the policy using random mCVRP instances with varying numbers of agents and customers and employ the trained policy without parameter changes to solve TSP, CVRP and mTSP to test its domain transferability.

## 3 METHOD

This section explains how the proposed model, given a state (a partial solution), assigns an idle vehicle to next node to visit under the sequential decision-making framework (see Figure 2).

### 3.1 STATE REPRESENTATION USING RELATIONSHIP-SPECIFIC MULTIPLE GRAPHS

The proposed model represents the global state $\mathbf{s}_t$ using as a weighted graph $\mathcal{G}_t = \mathcal{G}(V, E, w)$ where $V = \{V_V, V_C, V_R\}$, and $E$ is the set of edges between node $i, j \in V$ and $w$ is weight for edge $(i, j)$ (here, distance $d_{ij}$). Each node corresponding to vehicle, customer, and refueling station will be initialized with its associated states defined earlier.

Although we can assume that all nodes are connected with each other regardless of types and distance, we restrict the edge connection to its neighboring nodes to reduce the computational cost. Specifically, each type of node can define its connectivity range and connect an edge if any node is located within its range as follows (see Figure 2):

$$e_{vj} = 1 \quad \forall v \in V_v, \quad d(v, j) \leq R_V = f_t^v \tag{1}$$

$$e_{cj} = 1 \quad \forall c \in V_c, \quad d(c, j) \leq R_C \tag{2}$$

$$e_{rj} = 1 \quad \forall r \in V_R, \quad d(r, j) \leq R_R = \max_{v \in V_v} F^v \tag{3}$$

That is, the vehicle node $v \in V_v$ connects the edges with nodes that are located within its traveling distance (i.e., the current fuel level $f_t^v$). In addition, the customer nodes $c \in V_c$ connects the edges with nodes that are located within the constant range $R_C$. We set $R_C = 5$ while following a typical hyperparameter selection procedure. Finally, the refueling node $r \in V_R$ connects the edges with nodes within maximum distance that the vehicle with the largest fuel capacity can travel with the full loaded fuel (in this study $F^v = 10$ for all vehicles). Note that the target node $j$ that can be connected to each node can be any types of nodes.

## 3.2 NODE EMBEDDING USING GNN

The proposed model employs Graph neural network (GNN) (Scarselli et al., 2008) to compute the node embeddings for all nodes. The node embedding procedure starts with constructing the graph $\mathcal{G}_t$ out of the global state $\mathbf{s}_t = (\{s_t^v\}_{v=1}^{N_v}, \{s_t^c\}_{c=1}^{N_C}, \{s_t^r\}_{r=1}^{N_R})$. The method then compute the initial node embeddings $h_i$ and edge embeddings $h_{ij}$ for all the nodes and edges of $\mathcal{G}_t$ by employing encoder network. The sequel will explain how the GNN layer update these node and edge embeddings.

### 3.2.1 EDGE UPDATE

The edge feature $h_{ij}^\tau$ at $\tau$ iteration is updated using the edge updater function $\phi_E$ as

$$h_{ij}^\tau = \phi_E(h_{ij}^{\tau-1}, h_i^{\tau-1}, h_j^{\tau-1}), \quad \forall i \in V, \forall j \in \mathcal{N}_i \tag{4}$$

where $h_i^{\tau-1}$ and $h_j^{\tau-1}$ are node embedding vectors of node $i$ and node $j$ at $\tau - 1$ iteration.

### 3.2.2 EDGE FEATURE AGGREGATION

The updated edge feature $h_{ij}^\tau$ can be thought of as an message sent from node $j$ to node $i$. Node $i$ aggregates these messages from its neighboring nodes $j \in \{\mathcal{N}_V(i), \mathcal{N}_C(i), \mathcal{N}_R(i)\}$, where $\mathcal{N}_V(i)$ is the neighboring vehicle nodes of node $i$, as

$$(\bar{h}_{i,V}^\tau, \bar{h}_{i,C}^\tau, \bar{h}_{i,R}^\tau) = \left( \sum_{j \in \mathcal{N}_V(i)} \alpha_{ij} h_{ij}^\tau, \sum_{j \in \mathcal{N}_C(i)} \alpha_{ij} h_{ij}^\tau, \sum_{j \in \mathcal{N}_R(i)} \alpha_{ij} h_{ij}^\tau \right) \tag{5}$$

where the attention weight $\alpha_{ij} = \text{sofmax}_i(e_{ij})$, where $e_{ij} = f_e(s_i, s_j; w_e)$, scores the significance of node $j$ to node $i$. Note that message aggregation is separately conducted for different types of noes and the aggregated messages per type are concatenated.

### 3.2.3 NODE FEATURE UPDATE

The aggregated edge node embeddings $(\bar{h}_{i,V}^\tau, \bar{h}_{i,C}^\tau, \bar{h}_{i,R}^\tau)$ per its neighborhood type are then used to update the node embedding vector $h_i^\tau$ using node update function $\phi_V$ as

$$h_i^\tau = \phi_V \left( h_i^{\tau-1}, (\bar{h}_{i,V}^\tau, \bar{h}_{i,C}^\tau, \bar{h}_{i,R}^\tau) \right) \tag{6}$$

The node embedding procedure is repeated $H$ (hop) times for all nodes, and the final node embeddings $\mathbf{h}^H = \{h_i^H\}_{i=1}^N$ is used to determine the next assignment action of an idle vehicle.

## 3.3 DECISION MAKING USING NODE EMBEDDING

When an vehicle node $i$ reaches the assigned customer node, an event occurs and the vehicle node $i$ computes its node embedding $h_i^H$ and selects one of its feasible actions, choosing one of unvisited customer nodes or refueling nodes around vehicle node $i$. The probability for agent $i$ to choose node $j$, $a_t^i = j$, is computed by the parameterized actor $\pi(a_t^i = j | \mathbf{s}^t; \phi)$ as:

$$\pi(a_t^i = j | \mathbf{s}^t; \phi) = \frac{\exp(F(h_i^H, h_j^H; \phi)}{\sum_{k \in \mathcal{N}_V(i) \cup \mathcal{N}_R(i)} \exp(F(h_i^H, h_k^H; \phi)} \tag{7}$$

where $F(h_i^H, h_j^G; \phi)$ is the fitness function evaluating the goodness for agent $i$ to choose node $j$ as the next action (i.e., $a_t^i = j$).

### 3.4 TRAINING GRLTS

We train the proposed model using random mCVRP instances with varying numbers of vehicles, customers. We employ the actor-critic algorithm to train the parameters for the GNN, the critic, and the actor (GRLTS).

We first approximate the action-value function $Q^\pi(\mathbf{s}, \mathbf{a}) \approx \hat{Q}^\pi(\mathbf{s}, \mathbf{a}; \theta)$ using a neural network with the critic parameter $\theta$. The parameter $\theta$ for the centralized critic is optimized to minimize the loss $\mathcal{L}$:

$$\begin{aligned} \mathcal{L}(\theta) &= \mathbb{E}_{\mathbf{o}, \mathbf{a}, r, \mathbf{o}' \sim \mathcal{D}}[(\hat{Q}^\pi(\mathbf{s}, \mathbf{a}; \theta) - y)^2] \\ &= \mathbb{E}_{\mathbf{o}, \mathbf{a}, r, \mathbf{o}' \sim \mathcal{D}}[(\hat{Q}^\pi(\mathbf{h}^{N_{hop}}; \theta) - y)^2] \end{aligned} \tag{8}$$

where $y = r + \gamma \hat{Q}^{\pi'}(\mathbf{h}^{N_{hop}}; \theta')$ is the target value evaluated with the the actor $\pi' = \pi(\mathbf{a}|\mathbf{s}; \phi')$ and the critic $\hat{Q}^\pi(\mathbf{s}, \mathbf{a}; \theta')$ using the target parameters $\phi'$ and $\theta'$. In addition, $\mathcal{D}$ is a state transition memory.

To train the actor network, we use PPO method (Schulman et al., 2017) to maximize $J(\phi)$. PPO aims to maximize the clipped surrogate objective function as follow:

$$J^{\text{CLIP}}(\phi) = \mathbb{E}_t[\min(R_t(\phi)\hat{A}^t, \text{clip}(R_t(\phi), 1 - \epsilon, 1 + \epsilon)\hat{A}^t)] \tag{9}$$

where $R_t(\phi) = \frac{\pi_\phi(\mathbf{a}_t|\mathbf{s}_t)}{\pi_{\phi_{\text{old}}}(a^t|o^t)}$. We compute the advantage estimator $\hat{A}_t$ by running the policy for $T$ time steps as

$$\begin{aligned} \hat{A}_t &= \delta_t + \gamma\delta_{t+1} + \cdots + \cdots + \gamma^{T-1}\delta_{t+T-1} \\ \delta_t &= r_t + \gamma V(h_{t+1}^{N_{hop}}; \theta) - V(h_t^{N_{hop}}; \theta) \end{aligned} \tag{10}$$

In addition, equation (9) is added by a value function error and an entropy bonus for sufficient exploration because critic and actor share parameters as follow:

$$J_t^{\text{CLIP}'}(\phi) = \mathbb{E}_t[J_t^{\text{CLIP}}(\phi) - c_1(V_\theta(\mathbf{s}_t) - V_{\text{target}})^2 + c_2 H(\mathbf{s}_t, \pi_\phi(\cdot))] \tag{11}$$

where $V_\theta(\mathbf{s}_t) \approx V(\mathbf{h}_t^{N_{hop}}; \theta)$ from the centralized critic; $H$ denotes an entropy bonus; and $c_1$ and $c_2$ are hyperparameters.

On updating the centralized critic parameter $\theta$ and the decentralized (but, shared) actor parameter $\phi$, we use Monte-Carlo simulation. To sequentially update both parameters, we follow an update rule as follow:

$$\theta^{k+1}, \phi^{k+1} \leftarrow \arg\max_{\theta^i, \phi_k o^i, a^i \sim \pi_{\theta_k^i}} \mathbb{E}\left[J_t^{\text{CLIP}}(\phi) - c_1(V_\theta(\mathbf{s}_t) - V_{\text{target}})^2 + c_2 H(\mathbf{s}_t, \pi_\phi(\cdot))\right] \tag{12}$$

Algorithm (1) focus on sequence of the parameter update using above equations.

---

**Algorithm 1** Parameter update in decentralized actors with PPO and Monte-Carlo simulation

---

1: **for** Agent $i = 1, 2, \ldots, N$ **do**
2:     **for** $Update\ iteration = 1, 2, \ldots, K$ **do**
3:         Evaluate policy $\pi_{\theta_{old}}$ in environment for an experienced episode
4:         Compute $r_t(\phi)$ in Equation (9)
5:         Compute advantage estimates $\hat{A}_1, \ldots, \hat{A}_T$ in Equation (10)
6:         Optimize surrogate objective in Equation (11) with batch size $T$
7:         $\theta_{old} \leftarrow \theta, \phi_{old} \leftarrow \phi$ followed by (12)
8:     **end for**
9: **end for**

---

## 4 EXPERIMENTS

The trained policy is used to solve various types of vehicle routing problems, mCVRP, mTSP, CVRP, and TSP without changing the parameters. For each type except mCVRP, we use two types of data

sets: random instances with varying numbers of vehicles and customers whose locations are sampled randomly, and (2) benchmark problems obtained from the library (mTSP library, CVRP library, and TSP library).

To validate the proposed GRLTS, we develop mixed integer linear programming (MILP) formulations and compute the solutions using CPLEX 12.9 (all the optimization formulations are provided in the Appendix). We also use Google OR-Tools(Perron & Furnon) as the representative heuristic solvers. For the problems where other deep RL based approaches tried to solve (TSP and CVRP), we compare the performance of the proposed approach to those of deep RL baselines. All experiments are conducted on Windows 10, Intel(R) Core(TM) i9-9900K CPU 3.6 GHz, and 32GB RAM. GPU acceleration is not used in testing.

## 4.1 PERFORMANCE COMPARISON OF MCVRPS

Table 1: Performance comparison of mCVRPs on random instances

| Method | mCVRP25 | | | mCVRP50 | | | mCVRP100 | |
| --- | --- | --- | --- | --- | --- | --- | --- | --- |
| | $N_v = 2$ | $N_v = 3$ | $N_v = 5$ | $N_v = 2$ | $N_v = 3$ | $N_v = 5$ | $N_v = 5$ | $N_v = 10$ |
| ORT | **2.51** | **1.60** | 1.29 | 4.31 | 2.71 | 1.77 | - | **1.87** |
| | (4.5) | (16.7) | (45.2) | (12.1) | (19.2) | (51.9) | ($\infty$) | (489.2) |
| GRLTS | 3.15 | 2.0 | **1.20** | **3.72** | **2.65** | **1.71** | **2.71** | 1.88 |
| | (0.9) | (1.0) | (1.0) | (4.7) | (4.1) | (3.9) | (11.8) | (11.7) |
| Gap (%) | 25.5 | 25.0 | -7.0 | -13.7 | -2.2 | -3.4 | - | 0.5 |

Table 2: Scalability test of the trained GRLTS with large-scale mCVRPs

| mCVRP25-5 | mCVRP100-10 | mCVRP400-20 | mCVRP2500-50 |
| --- | --- | --- | --- |
| 1.2 (0.98) | 1.88 (12.1) | 4.09 (420.2) | 11.71 (3,861.2) |

We apply the trained policy (GRLTS) to solve mCVRP with different numbers of vehicles $N_v \in 2, 3, 5$ and the numbers of customers $N_c \in 25, 50, 100$. The number of refueling stations are set to be 4, 5 and 10 in case of $N_c = 25$, $N_c = 50$ and $N_c = 100$. For every combination of $N_v$ and $N_c$, we randomly generate 100 mCVRP instances, each of which has randomly located customer nodes and refueling nodes. The x and y coordinates of each node is randomly sampled from the uniform distributions; $x \sim U[0, 1]$ and $y \sim U[0, 1]$, respectively. We employ GRLTS and ORT to solve the same 100 random mCVRP instances and compute the average makespan and the computational time required to solve the mCVRP instance with the policies. Table 1 compares the average makespan and computational times.

ORT (Google OR-tool) produces the best results for the small-sized problems with reasonable computational time; however, it requires extensive time or even fails to compute any feasible solution for the large scale problem ($\infty$ means that ORT cannot find a feasible solution). Notably, the GRLTS achieves a better performance than ORT for large-scale problems with a significantly shorter computational time.

To validate the scalability of the proposed model, we further conduct test experiments with the four cases: $N_c = 25$ with $N_v = 5$, $N_c = 100$ with $N_v = 10$, $N_c = 400$ with $N_v = 20$, and $N_c = 2,500$ with $N_v = 50$. Table 2 shows how the makespan and the computational time increases with the size of the problem. By comparing the ratio between the number of customers that each vehicle need to serve, (5: 10: 20: 50), and the makespan , (1.20: 1.88: 4.09: 11.71), we can roughly confirms that the trained model can perform reasonably well even in large sized problem that have never been experience during the training.

For all different sizes of the mCVRP problem, we also compare the performance of the proposed method with the solution computed by CPLEX solver. Because it typically takes a long time to compute the near-optimum solution by CPLEX, we only compare the single mCVRP case. The result is provided in the Appendix.

## 4.2 Performance comparison on mTSP

We apply the trained network (without parameter changes) to solve mTSP, which seeks to minimize the total completion time of multiple vehicles (minmax mTSP). This problem is a relaxed version of mCVRP in that it does not require the capacity constraints (we maintains the fuel level to be maximum during execution). We solve the randomly generated 100 mTSP instances for every combination of $N_c \in 50, 100, 400$ and $N_v \in 2, 4, 6, 8$ (or $N_v \in 10, 20, 30, 40$). Table 7 compares the average makespan and computational time. The trained model outperforms ORT in the large-size problems (see $N_c = 100$ and $N_c = 400$ cases in Table 7) in terms of both makespan and the computational time. For large scale problems, GRLTS achieves roughly 28% shorter makespan than ORT on average with significantly reduced computational time (50% reduction).

Table 3: Performance comparison on mTSP for random instances

| | mTSP50 | | | | mTSP100 | | | | mTSP400 | | | |
|---|---|---|---|---|---|---|---|---|---|---|---|---|
| $N_v$ | 2 | 4 | 6 | 8 | 2 | 4 | 6 | 8 | 10 | 20 | 30 | 40 |
| ORT | 3.20 | 2.21 | 1.60 | 1.11 | 5.13 | 2.69 | 2.25 | 1.71 | 3.02 | 2.0 | 1.91 | 1.61 |
| | (1.3) | (1.2) | (1.2) | (1.2) | (25.4) | (21.1) | (21.5) | (22.1) | (1358.8) | (965.9) | (1115.6) | (1025.7) |
| GRLTS | 3.94 | 2.83 | 1.47 | 0.98 | 5.13 | 2.34 | 2.16 | 1.71 | 2.64 | 2.06 | 1.34 | 1.15 |
| | (3.1) | (3.2) | (3.0) | (3.0) | (16.3) | (11.2) | (10.9) | (11.1) | (432.6) | (452.1) | (442.1) | (465.7) |
| Gap (%) | 23.1 | 28.1 | -8.1 | -11.7 | 0.0 | -13.0 | -4.0 | 0.0 | -12.6 | 3.0 | -29.8 | -28.6 |

Similarly, for all the sizes of mTSP case, we compare the performance of GRLTS to the solution computed by CPLEX in the Appendix. We also employed the trained model to solve mTSP benchmark problems in MTSPLib [1] solving MinMax mTSP. The performance results on 124 instances (four different number of vehicles per 31 different maps) are provided in Table 9 in Appendix.

## 4.3 Performance comparison on CVRP (payload capacity)

Table 4: Performance comparison on CVRP for random instances

| Methods | CVRP20 | | | CVRP50 | | | CVRP100 | | |
|---|---|---|---|---|---|---|---|---|---|
| | Obj. | Gap (%) | Time | Obj. | Gap (%) | Time | Obj. | Gap (%) | Time |
| L2I (Lu et al., 2019) | 6.12 | - | 12m | 10.35 | - | 17m | 15.57 | - | 24m |
| LKH3 (Helsgaun, 2017) | 6.14 | 0.33 | 2h | 10.38 | 0.29 | 7h | 15.64 | 0.45 | 13h |
| OR-Tools | 6.46 | 5.56 | 2m | 11.27 | 8.89 | 13m | 17.12 | 9.96 | 46m |
| AM (Kool et al., 2018) greedy | 6.4 | 4.58 | 1s | 10.61 | 2.51 | 3s | 16.17 | 3.85 | 8s |
| AM Sampling | 6.25 | 2.12 | 6m | 10.59 | 2.32 | 28m | 16.12 | 3.53 | 2h |
| Nazari et al. (2018) | 6.4 | 4.58 | | 11.15 | 7.73 | | 16.96 | 8.93 | |
| Chen & Tian (2019) | 6.16 | 0.65 | | 10.51 | 1.55 | | 16.1 | 3.40 | |
| Random Sweep | 7.08 | 15.69 | | 12.96 | 25.22 | | 20.33 | 30.57 | |
| Random CW | 6.81 | 11.27 | | 12.25 | 18.36 | | 18.96 | 21.77 | |
| GRLTS (Ours) | 6.97 | 13.89 | 2s | 12.91 | 24.73 | 5s | 19.74 | 26.78 | 15s |

We employ the trained model to solve randomly generated 100 CVRP instances with $N_c = 20, 50, 100$ with a single capacitated vehicle. All nodes are randomly scattered in the unit square of $[0, 1] \times [0, 1]$. To make the trained policy compatible with the CVRP settings, we fix the number of vehicles to be one and use a single refueling node as if it were the depot in CVRP environment. As a result, the single vehicle needs to revisit the depot due to payload capacity (which is equivalent to fuel constraint). The payload capacity is set to be 30, 40 and 50 for CVRP20, CVRP50 and CVRP100, respectively. Demand are uniformly sampled from $\{1, \ldots, 9\}$. These test cases have been widely used by the studies seeking to develop RL-based solvers for CVRP problems.

Table 4 summaries results for the random CVRP environment. L2I and LKH3 are well-known best performing heuristic algorithms developed from the OR community, thus can serve as an oracle for comparing the performance. We also consider other RL-based approaches (Kool et al., 2018;

---

[1] www.infoiasi.ro/ mtsplib

Nazari et al., 2018; Chen & Tian, 2019). In general, our model is not outperforming other RL-based approaches and OR-tool. However, all RL-based approaches, except our model, are trained under the same environment of CVRP cases (the training and test instances are sampled from distribution). However, our model is trained with the complete different mCVRP environment and tested with the CVRP environment (strong transferability).

We also employed the trained policy to solve the CVRP benchmark problem instances from CVRPLib (Uchoa et al., 2017). Table 10 in Appendix provides all the performance results compared with other RL-baseline models. The results show that our model has better performance than one of the state-of-art RL-based approach (Kool et al., 2018).

## 4.4 PERFORMANCE COMPARISON OF TSP

Table 5: Performance comparison on TSP for random instances

| Methods | CVRP20 | | | CVRP50 | | | CVRP100 | | |
|---|---|---|---|---|---|---|---|---|---|
| | Obj. | Gap (%) | Time | Obj. | Gap (%) | Time | Obj. | Gap (%) | Time |
| Concorde | 3.84 | 0.00 | 1m | 5.70 | 0.00 | 2m | 7.76 | 0 | 3m |
| LKH3 | 3.84 | 0.00 | 18s | 5.70 | 0.00 | 5m | 7.76 | 0 | 21m |
| OR-Tools | 3.85 | 0.26 | 0s | 5.80 | 1.75 | 1s | 7.99 | 2.96 | 3s |
| AM (greedy) | 3.85 | 0.26 | 0s | 5.80 | 1.75 | 2s | 8.12 | 4.64 | 6s |
| AM (sampling) | 3.84 | 0.00 | 5m | 5.73 | 0.53 | 24m | 7.94 | 2.32 | 1h |
| Nearest Insertion | 4.33 | 12.76 | 1s | 6.78 | 18.95 | 2s | 9.46 | 21.91 | 6s |
| Random Insertion | 4.00 | 4.17 | 0s | 6.13 | 7.54 | 1s | 8.52 | 9.79 | 3s |
| Farthest Insertion | 3.93 | 2.34 | 1s | 6.01 | 5.44 | 2s | 8.35 | 7.60 | 7s |
| Bello et al. (2016) | 3.89 | 1.30 | | 5.95 | 4.39 | | 8.3 | 6.96 | |
| Khalil et al. (2017) | 3.89 | 1.30 | | 5.99 | 5.09 | | 8.31 | 7.09 | |
| GRLTS (Ours) | 3.92 | 2.08 | 1s | 6.32 | 10.88 | 3s | 8.79 | 13.27 | 9s |

Lastly, we employ the trained model to solve 100 randomly generated TSP with different number of customers $N_c = 20, 50, 100$ and a single vehicle. Table 5 shows summarized the results. Although GRLTS is not outperforming other RL-based scheduling methods, it shows the reasonably performance that is comparable to some of well known heuristic algorithms. Given that the GRLTS is trained with mCVRP environment and have never seen TSP instances, this result can validate that GRLTS can be transferred to TSP as well.

We also employed the trained policy to solve the TSP benchmark problem instances fromTSPLib (Reinhelt, 2014). Table 8 in Appendix provides all the performance results compared with other RL-baseline models. The results show that our model has comparable performance with the-state-of-art RL-based approaches (GPN (Ma et al., 2019) and S2V-DQN (Khalil et al., 2017)) and heuristic algorithms, especially for large-scale TSP problems.

## 5 CONCLUSION

We proposed the Graph-centric RL-based Transferable Scheduler for various vehicle routing problems using graph-centric state presentation (GRLTS) that can solve any types of vehicle routing problems such as mCVRP, mTSP, CVRP, and TSP. The transferability is achieved by graph-centric representation learning that can generalize well over various relationships among vehicles, customers, and refuel stations (depot). GRLTR is computationally efficient for solving very large-scale vehicle routing problems with complex constraints, which provides potential that such RL-based scheduler can be used for large-scale realistic applications in logistics, transportation, and manufacturing.

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

## A APPENDIX

### A.1 MILP FORMULATIONS FOR ROUTING PROBLEMS

The MILP formulations for mCVRP, mTSP, and TSP are provided in this section. All problems are formulated to find the optimum routes of vehicles to minimize the makespan. For TSP formulation, the makespan minimization is equal to total distance minimization due to single-vehicle operation. Although mTSP here is same as multi-VRP by its definition, we denote the problem as mTSP because we focus on the makespan minimization problem with multi-vehicle operation.

#### A.1.1 TSP

A fundamental formulation in the context of routing problems is based on TSP formulation by Miller-Tucker-Zemlin (MTZ) formulation (Miller et al., 1960):

$$\text{minimize} \quad \sum_{i \in V} \sum_{j \in V, i \neq j} d_{ij} x_{ij}$$

$$
\begin{aligned}
\text{subject to.} \quad & \sum_{i \in V} x_{ij} = 1, & & \forall j \in V : i \neq j, & (1) \\
& \sum_{j \in V} x_{ij} = 1, & & \forall i \in V : i \neq j, & (2) \\
& u_i - u_j + |V| x_{ij} \leq |V| - 1, & & \forall i, j \in V \setminus vStart, & (3) \\
& 0 \leq u_i \leq |V| - 1, & & \forall i \in V \setminus vStart, & (4) \\
& x_{ij} \in \{0, 1\}, & & \forall i, j \in V, & (5) \\
& u_i \in \mathbb{Z}, & & \forall i \in V \setminus vStart. & (6)
\end{aligned}
$$

The routing problem is defined in a graph $G(V, E, w)$ where $V$, $E$ and $w$ are nodes, edges, and weight (distance $d$ or cost $c$). $x_{ij}$ is 1 if the edge between node $i$ and node $j$ exists, otherwise, 0. Here, corresponding distance of the edge is $d_{ij}$. Constraint (3,4) with dummy variable $u_i$ is for subtour elimination. As setting $d_{ij} = 0, \forall j \in vStart$, the problem can be arbitrary end assumption.

#### A.1.2 MTSP

The goal of mTSP is to find the sub-routes of multiple vehicles to minimize the makespan. Thus, decision variables are expanded to multi-vehicle settings, and the objective function is modified from A.1.1 to consider the makespan minimization setting (MinMax problem).

$$\text{minimize} \quad Q$$

$$
\begin{aligned}
\text{subject to.} \quad & \sum_{i \in V} \sum_{j \in V} d_{ij} x_{aij} \leq Q, & & \forall a \in A : i \neq j, & (1) \\
& \sum_{j \in V} x_{aij} = 1, & & \forall a \in A, i \in vStart_a : i \neq j, & (2) \\
& \sum_{a \in A} \sum_{i \in G} \sum_{j \in G} x_{aij} = 1, & & \forall j \in vTask : i \neq j, & (3) \\
& \sum_{i \in G} x_{aij} - \sum_{i \in G} x_{aji} = 0, & & \forall a \in A, j \in V : i \neq j, & (4) \\
& u_{ai} - u_{aj} + |V| x_{aij} \leq |V| - 1, & & \forall a \in A, j \in V \setminus vStart_a : i \neq j, & (5) \\
& 0 \leq u_{ai} \leq |V| - 1, & & \forall a \in A, i \in V \setminus vStart_a, & (6) \\
& x_{aij} \in \{0, 1\}, & & \forall a \in A, \forall i, j \in V, & (7) \\
& u_{ai} \in \mathbb{Z}, & & \forall a \in A, i \in V \setminus vStart_a. & (8)
\end{aligned}
$$

Here, we denoted $Q$ as the makespan, which is the longest traveling distance among multiple vehicles. By minimizing this maximum traveling distance, the above formulation can minimize the makespan. Therefore, the above formulation is to minimize the makespan. We also allow a vehicle to start a tour at any staring node $vStart_a$ (constraint (2)).

A.1.3  MCVRP

We extend the above mTSP formulation in A.1.2 to include the fuel constraint (i.e., allowable traveling distance per sub-tour). In this formulation, vehicles can start their tour at an arbitrary location.

$$
\begin{aligned}
\text{minimize} \quad & Q \\
\text{subject to.} \quad & \sum_{i \in V} \sum_{j \in V} d_{ij} x_{aij} \leq Q, & \forall a \in A : i \neq j, & \quad (1) \\
& \sum_{j \in V} x_{aij} = 1, & \forall a \in A, i \in vStart_a : i \neq j, & \quad (2) \\
& \sum_{a \in A} \sum_{i \in G} \sum_{j \in G} x_{aij} = 1, & \forall j \in vTask : i \neq j, & \quad (3) \\
& \sum_{i \in G} x_{aij} - \sum_{i \in G} x_{aji} = 0, & \forall a \in A, j \in V : i \neq j, & \quad (4) \\
& u_{ai} - u_{aj} + |V| x_{aij} \leq |V| - 1, & \forall a \in A, j \in V \setminus vStart_a : i \neq j, & \quad (5) \\
& 0 \leq u_{ai} \leq |V| - 1, & \forall a \in A, i \in V \setminus vStart_a, & \quad (6) \\
& x_{aij} \in \{0, 1\}, & \forall a \in A, \forall i, j \in V, & \quad (7) \\
& u_{ai} \in \mathbb{Z}, & \forall a \in A, i \in V \setminus vStart_a & \quad (8) \\
& f_{ai} = F_a, & \forall a \in A, i \in vRefuel, & \quad (9) \\
& f_{ai} - d_{ij} x_{aij} \geq 0, & \forall a \in A, \forall i, j \in vRefuel, & \quad (10) \\
& f_{aj} - f_{ai} + d_{ij} x_{aij} \geq F_a (1 - x_{aij}), & \forall a \in A, \forall i, j \in vTask. & \quad (11)
\end{aligned}
$$

Constraint (9) indicates that a vehicle can charge fuel as much as the vehicle's maximum fuel capacity $F_a$. In addition, constraint (10) requires that a vehicle must have enough remaining fuel to visit a refueling node. Constraint (11) indicates that a vehicle cannot consume (travel) more than its fuel capacity. We allow vehicles to visit refueling nodes as many times as possible by introducing a sufficient number of dummy variables.

A.1.4  CVRP

Similar to the above mCVRP with fuel constraint, we also solve CVRP constrained by payload capacity (not fuel constraint) and with single vehicle ($m = 1$ so that $|A| = 1$). We consider a single depot to start a mission and unload some burdens. Therefore, $vRefuel$ in the MILP of mCVRP becomes same as $vStart_a$. Fuel consumption as much as distance $d_{ij}$ between node $i$ and node $j$ is equivalent to payload capacity consumption as much as demand $d_j$ at node $j$. As a result, constraint (9-11) of the mCVRP model becomes:

$$
\begin{aligned}
& c_{ai} = C_a, & \forall a \in A, i \in vStart, & \quad (9) \\
& c_{aj} - c_{ai} + d_j x_{aij} \geq C_a (1 - x_{aij}), & \forall a \in A, \forall i, j \in vTask. & \quad (10)
\end{aligned}
$$

The constraint (10) in the mCVRP model is dropped because payload capacity is independent in returning to the starting depot.

A.2  COMPARISON WITH CPLEX SOLUTIONS

For a single mCVRP and mTSP instances, we compare the performance of the proposed approach with the exact solution computed by CPLEX. This experiment shows how the solution computed from the proposed method is comparable to the near-optimum solution calculated by the powerful optimization solver. Note that the mCVRP instance for this experiment was generated from the grid environment. The distance between customer nodes are different from the experiments for Table 1.

## A.2.1 MCVRP

Table 6: Performance comparison of mCVRPs on random instances

| $N_v$ | $N_c = 25$ | | | $N_v$ | $N_c = 50$ | | | $N_v$ | $N_c = 100$ | | |
| | CPX | ORT | RL | | CPX | ORT | RL | | CPX | ORT | RL |
|---|---|---|---|---|---|---|---|---|---|---|---|
| 2 | 20 ($\infty$) | **15** (3.9) | 20 (0.96) | 2 | - | 34 (38.3) | **30** (4.8) | 5 | - | **27** (510.3) | 29(11.9) |
| 3 | 14 ($\infty$) | **10** (36.6) | 16 (1.01) | 3 | - | **22** (2.2) | **22** (4.1) | 10 | - | - | **20** (12.1) |
| 5 | 12 ($\infty$) | - | **8** (0.98) | 5 | - | **14** (1.7) | **14** (3.8) | | | | |

We apply the already trained model to solve mCVRP with different numbers of vehicles $N_v \in 2, 3, 5$ and the numbers of customers $N_c \in 25, 50, 100$. The number of refueling stations are set to be 4, 5 and 10 in case of $N_c = 25$, $N_c = 50$ and $N_c = 100$. For all cases, we compute the total completion time and the computational time (the number in the parenthesis) required to construct a scheduling. We set the limit of the computational time to be 18,000 (sec) for all cases. The symbol $\infty$ indicate the case where the computational time is reached and the solution at that moment is used in the table. The blanks with the hyphen (-) indicate the case where the algorithm could not find any feasible solution.

First, CPX (CPLEX) can only solve for mCVRPs with $N_c = 25$; for large scale problems, it cannot compute any feasible solution. ORT (Google OR-tool) produces the best results for the small sized problem with reasonable computational time; however, for the large scale problem it requires extensive amount of time or even fail to compute any feasible solution. Notably, the proposed GRLTS achieves the good performance with the least amount of computational time.

## A.2.2 MTSP

For the grid environment, a single mTSP instance was generated and used for comparing the performances of GRLTS to CPLEX and ORT.

Table 7: Performance comparison on mTSP for random instances

| $N_v$ | $N_c = 49$ | | | $N_v$ | $N_c = 100$ | | | $N_v$ | $N_c = 400$ | | |
| | CPX | ORT | RL | | CPX | ORT | RL | | CPX | ORT | RL |
|---|---|---|---|---|---|---|---|---|---|---|---|
| 2 | **24*** (7.9) | 26 (2.1) | 32 (3.0) | 2 | 51 ($\infty$) | **57** (32.2) | **57** (15.9) | 10 | - | 63 (1619.8) | **55** (361.2) |
| 4 | **13** ($\infty$) | 18 (1.1) | 23 (3.2) | 4 | 54 ($\infty$) | 31 (16.9) | **26** (10.7) | 20 | - | **41** (782.8) | 43 (415.9) |
| 6 | 13 ($\infty$) | 13 (1.1) | **12** (3.3) | 6 | 46 ($\infty$) | 25 (15.4) | **24** (10.5) | 30 | - | 44 (1192.1) | **28** (452.3) |
| 8 | **7*** (6467.3) | 9 (1.6) | 8 (3.4) | 8 | 51 ($\infty$) | 19 (13.8) | **19** (10.8) | 40 | - | 39 (627.0) | **24** (485.1) |

## A.3 EXPERIMENT RESULTS ON BENCHMARK PROBLEMS

We apply the trained network to solve benchmark problems for TSP, mTSP and CVRP.

## A.3.1 TSP

Table 8 shows results of TSPLib (Reinhelt, 2014) with some RL-based approaches (GPN (Ma et al., 2019) and S2V-DQN (Khalil et al., 2017) and heuristics. The average performance of the proposed model is not outperforming the state of the art RL approaches. However, we observed that the proposed method is scalable in that the performance on the large-scale problem does not degrade compared to other approaches. We compare the performance by problem instance size (eil51 $\sim$ tsp225 Vs. pr226 $\sim$ pch442). Our model showed 1.1% reduction in the optimality gap (15.4% $\rightarrow$ 14.3%). However, the other RL-based approaches show much higher increases in the optimality gap (Drori et al. (2020): 3.1% $\rightarrow$ 10.0%, GPN: 14.1% $\rightarrow$ 33.1%, Drori et al. (2020): 4.7% $\rightarrow$ 10.6%).

Table 8: Performance comparison on TSP library

| City | Opt. | RL | | | | Approx. | | |
|---|---|---|---|---|---|---|---|---|
| | | Ours | Drori et al. (2020) | GPN | S2V-DQN | Farthest | 2-opt | Nearest |
| eil51 | 426 | 471 | 439 | 485 | 439 | 448 | 452 | 514 |
| berlin52 | 7542 | 8269 | 7681 | 8795 | 7734 | 8121 | 7778 | 8981 |
| st70 | 675 | 770 | 684 | 701 | 685 | 729 | 701 | 806 |
| eil76 | 538 | 648 | 555 | 591 | 558 | 583 | 597 | 712 |
| pr76 | 108159 | 135145 | 112699 | 118032 | 111141 | 119649 | 125276 | 153462 |
| rat99 | 1211 | 1368 | 1268 | 1472 | 1250 | 1319 | 1351 | 1565 |
| kroA100 | 21282 | 25282 | 21452 | 24806 | 22335 | 23374 | 23306 | 26856 |
| kroB100 | 22141 | 25939 | 22488 | 24369 | 22548 | 24035 | 23129 | 29155 |
| kroC100 | 20749 | 23644 | 21427 | 24780 | 21468 | 21818 | 22313 | 26327 |
| kroD100 | 21294 | 23397 | 21555 | 23494 | 21886 | 22361 | 22754 | 26950 |
| kroE100 | 22068 | 26326 | 22267 | 23467 | 22820 | 23604 | 25325 | 27587 |
| rd100 | 7910 | 9237 | 8243 | 8844 | 8305 | 8652 | 8832 | 9941 |
| eil101 | 629 | 753 | 650 | 704 | 667 | 687 | 694 | 825 |
| lin105 | 14379 | 17173 | 14571 | 15795 | 14895 | 15196 | 16184 | 20363 |
| pr124 | 59030 | 66554 | 59729 | 67901 | 61101 | 61645 | 61595 | 69299 |
| bier127 | 118282 | 125233 | 120672 | 134089 | 123371 | 127795 | 136058 | 129346 |
| ch130 | 6110 | 6987 | 6208 | 6457 | 6361 | 6655 | 6667 | 7575 |
| pr136 | 96772 | 111563 | 98957 | 110790 | 100185 | 104687 | 103731 | 120778 |
| pr144 | 58537 | 59197 | 60492 | 67211 | 59836 | 62059 | 62385 | 61651 |
| ch150 | 6528 | 7264 | 6729 | 7074 | 6913 | 6866 | 7439 | 8195 |
| kroA150 | 26528 | 32492 | 27419 | 30260 | 28076 | 28789 | 28313 | 33610 |
| kroB150 | 26130 | 32438 | 27165 | 29141 | 26963 | 28156 | 28603 | 32825 |
| u159 | 42080 | 51542 | 43687 | 52642 | 45620 | 46842 | 42976 | 53637 |
| rat195 | 2323 | 2774 | 2384 | 2686 | 2567 | 2620 | 2569 | 2762 |
| d198 | 15780 | 16217 | 17754 | 19249 | 16855 | 16161 | 16705 | 18830 |
| kroA200 | 29368 | 34160 | 30553 | 34315 | 30732 | 31450 | 32378 | 35798 |
| kroB200 | 29437 | 35848 | 30381 | 33854 | 31910 | 31656 | 32853 | 36982 |
| ts225 | 126643 | 141754 | 130493 | 147092 | 140088 | 140625 | 143197 | 152494 |
| tsp225 | 3916 | 4536 | 4091 | 4988 | 4219 | 4233 | 4046 | 4746 |
| pr226 | 80369 | 83861 | 86438 | 85186 | 82869 | 84133 | 85306 | 94390 |
| gil262 | 2378 | 2889 | 2523 | 5554 | 2539 | 2638 | 2630 | 3218 |
| pr264 | 49135 | 51836 | 52838 | 67588 | 53790 | 54954 | 58115 | 58634 |
| a280 | 2579 | 3059 | 2742 | 3019 | 3007 | 3011 | 2775 | 3311 |
| pr299 | 48191 | 59257 | 53371 | 68011 | 55413 | 52110 | 52058 | 61252 |
| lin318 | 42029 | 49167 | 45115 | 47854 | 45420 | 45930 | 45945 | 54034 |
| rd400 | 15281 | 16631 | 16730 | 17564 | 16850 | 16864 | 16685 | 19168 |
| fl417 | 11861 | 12768 | 13300 | 14684 | 12535 | 12589 | 12879 | 15288 |
| pr439 | 107217 | 124333 | 126849 | 137341 | 122468 | 122899 | 111819 | 131258 |
| pch442 | 50778 | 61757 | 55750 | 58352 | 59241 | 57149 | 57684 | 60242 |
| Gap | 0 | 15.1 | 4.9 | 19.0 | 6.2 | 8.4 | 9.1 | 24.7 |

### A.3.2   mTSP

Table 9 shows results from mTSPLib [2] solving MinMax TSP. Our model shows 3% longer results than ORT. However, computational time is, on average, about 30 seconds faster, which is about 45%. In addition, we calculate difference of the optimality gap by $N_v = 2, 3, 5, 7$. Our model shows better performance as $N_v$ increases ($N_v = 2$ case: 9.8%, $N_v = 3$ case: 8.9%, $N_v = 5$ case: 2.8%, $N_v = 7$ case: -7.5%)

---

[2] www.infoiasi.ro/ mtsplib

Table 9: Performance comparison on mTSP library

| Maps | $N_v$ | ORT MS | ORT Time | Ours MS | Ours Time | gap | Maps | $N_v$ | ORT MS | ORT Time | Ours MS | Ours Time | gap |
|---|---|---|---|---|---|---|---|---|---|---|---|---|---|
| eil51 | 2 | 223 | 1.14 | 281 | 2.95 | 26.2 | kroB100 | 2 | 12214 | 15.06 | 13721 | 9.41 | 12.3 |
| | 3 | 159 | 0.67 | 244 | 2.92 | 53.7 | | 3 | 8957 | 14.81 | 10197 | 9.52 | 13.8 |
| | 5 | 120 | 1.48 | 165 | 2.93 | 37.5 | | 5 | 7108 | 10.31 | 6271 | 9.19 | -13.3 |
| | 7 | 109 | 1.03 | 86 | 2.67 | -26.5 | | 7 | 7108 | 10.24 | 6119 | 9.49 | -16.2 |
| berlin52 | 2 | 4634 | 1.27 | 4247 | 3.01 | -9.1 | kroC100 | 2 | 11440 | 8.90 | 12453 | 9.39 | 8.9 |
| | 3 | 3195 | 1.35 | 3452 | 3.03 | 8.0 | | 3 | 8725 | 12.58 | 7985 | 9.48 | -9.3 |
| | 5 | 2606 | 1.18 | 2179 | 3.03 | -19.6 | | 5 | 6616 | 15.66 | 6291 | 9.50 | -5.2 |
| | 7 | 2440 | 1.42 | 2182 | 2.87 | -11.8 | | 7 | 6154 | 12.83 | 5740 | 9.30 | -7.2 |
| eil76 | 2 | 288 | 4.46 | 361 | 5.81 | 25.4 | kroD100 | 2 | 13130 | 11.27 | 12815 | 9.32 | -2.5 |
| | 3 | 212 | 3.26 | 227 | 5.75 | 7.2 | | 3 | 8889 | 9.36 | 9359 | 9.40 | 5.3 |
| | 5 | 179 | 4.79 | 170 | 5.61 | -5.1 | | 5 | 6976 | 9.67 | 6660 | 9.34 | -4.7 |
| | 7 | 179 | 4.78 | 121 | 5.41 | -47.4 | | 7 | 6485 | 12.00 | 7246 | 9.25 | 11.7 |
| st70 | 2 | 388 | 3.77 | 451 | 4.87 | 16.2 | kroE100 | 2 | 13424 | 9.02 | 13611 | 9.53 | 1.4 |
| | 3 | 285 | 2.04 | 282 | 4.98 | -1.2 | | 3 | 9334 | 14.61 | 8929 | 10.76 | -4.5 |
| | 5 | 251 | 2.26 | 249 | 4.69 | 0.8 | | 5 | 7599 | 20.27 | 6992 | 10.05 | -8.6 |
| | 7 | 251 | 2.22 | 253 | 4.78 | 0.9 | | 7 | 8727 | 17.14 | 5868 | 9.42 | -18.7 |
| pr76 | 2 | 60679 | 4.99 | 80789 | 5.69 | 33.1 | eil101 | 2 | 345 | 11.23 | 386 | 9.70 | 11.8 |
| | 3 | 51074 | 6.25 | 65567 | 5.67 | 28.4 | | 3 | 237 | 10.32 | 264 | 9.64 | 11.3 |
| | 5 | 39167 | 3.55 | 48143 | 5.48 | 22.9 | | 5 | 163 | 11.15 | 167 | 9.67 | 2.3 |
| | 7 | 38116 | 5.85 | 33082 | 5.50 | -15.2 | | 7 | 123 | 15.29 | 141 | 9.63 | 14.6 |
| rat99 | 2 | 709 | 16.47 | 756 | 9.11 | 6.6 | lin105 | 2 | 8846 | 18.50 | 10220 | 10.20 | 15.5 |
| | 3 | 549 | 13.86 | 565 | 9.28 | 2.9 | | 3 | 7112 | 16.20 | 8168 | 10.30 | 14.8 |
| | 5 | 463 | 11.12 | 373 | 9.14 | -24.0 | | 5 | 7060 | 18.79 | 5487 | 10.34 | -28.7 |
| | 7 | 440 | 12.01 | 367 | 9.24 | -19.8 | | 7 | 6444 | 18.84 | 4802 | 10.35 | -34.2 |
| rd100 | 2 | 4406 | 17.94 | 4520 | 9.54 | 2.6 | pr107 | 2 | 31692 | 14.29 | 26369 | 10.96 | 20.2 |
| | 3 | 3224 | 20.31 | 3136 | 9.61 | -2.8 | | 3 | 24904 | 12.82 | 23283 | 10.90 | -7.0 |
| | 5 | 2435 | 20.78 | 2629 | 9.32 | 8.0 | | 5 | 21455 | 16.70 | 20985 | 10.80 | -2.2 |
| | 7 | 2803 | 17.38 | 2375 | 9.38 | -18.0 | | 7 | 21365 | 17.45 | 20620 | 10.81 | -3.6 |
| kroA100 | 2 | 12647 | 12.19 | 13560 | 9.49 | 7.2 | pr124 | 2 | 35094 | 18.82 | 36472 | 15.02 | 3.9 |
| | 3 | 9116 | 14.37 | 9635 | 9.49 | 5.7 | | 3 | 27257 | 16.70 | 27598 | 14.75 | 1.3 |
| | 5 | 8229 | 13.11 | 7462 | 9.42 | -10.0 | | 5 | 23231 | 21.44 | 23651 | 14.75 | 1.8 |
| | 7 | 8229 | 13.31 | 7147 | 9.24 | -15.1 | | 7 | 22725 | 25.29 | 21526 | 14.75 | -5.6 |
| bier127 | 2 | 62458 | 22.72 | 67706 | 15.83 | 8.4 | u159 | 2 | 26171 | 47.80 | 27864 | 26.35 | 6.5 |
| | 3 | 47347 | 22.61 | 52749 | 15.68 | 11.4 | | 3 | 19014 | 56.26 | 19519 | 25.87 | 2.7 |
| | 5 | 29560 | 38.21 | 30946 | 15.76 | 4.7 | | 5 | 14308 | 72.44 | 16719 | 26.34 | 16.9 |
| | 7 | 24891 | 47.49 | 25074 | 15.48 | 0.7 | | 7 | 14776 | 67.59 | 16077 | 26.03 | 8.8 |
| ch130 | 2 | 3393 | 31.44 | 3755 | 16.85 | 10.7 | rat195 | 2 | 1285 | 205.86 | 1524 | 43.66 | 18.6 |
| | 3 | 2550 | 23.90 | 3425 | 16.65 | 34.3 | | 3 | 975 | 101.40 | 937 | 43.82 | -4.0 |
| | 5 | 1575 | 39.03 | 2307 | 16.69 | 46.5 | | 5 | 708 | 152.43 | 638 | 43.57 | -10.9 |
| | 7 | 1338 | 32.88 | 2077 | 16.30 | 55.2 | | 7 | 631 | 130.31 | 495 | 43.85 | -27.5 |
| pr136 | 2 | 59999 | 34.54 | 59113 | 18.18 | -1.5 | d198 | 2 | 10416 | 170.17 | 9226 | 45.20 | -12.9 |
| | 3 | 40643 | 14.45 | 45017 | 18.25 | 10.8 | | 3 | 9663 | 71.48 | 7813 | 45.36 | -23.7 |
| | 5 | 29958 | 29.59 | 32858 | 18.14 | 9.7 | | 5 | 8577 | 113.45 | 6086 | 45.21 | -40.9 |
| | 7 | 30138 | 20.66 | 30222 | 17.98 | 0.2 | | 7 | 8577 | 115.69 | 5892 | 45.64 | -45.6 |
| pr144 | 2 | 38104 | 23.11 | 52344 | 20.72 | 37.4 | kroA200 | 2 | 17148 | 154.68 | 20696 | 46.46 | 20.7 |
| | 3 | 31305 | 36.20 | 42179 | 20.72 | 34.7 | | 3 | 12253 | 161.74 | 13778 | 46.44 | 12.4 |
| | 5 | 25262 | 46.69 | 28153 | 20.55 | 1.4 | | 5 | 8185 | 190.01 | 10770 | 46.33 | 31.6 |
| | 7 | 24361 | 42.25 | 27168 | 20.52 | 11.5 | | 7 | 6675 | 186.54 | 7142 | 46.36 | 7.0 |
| ch150 | 2 | 3914 | 43.40 | 3637 | 22.75 | -7.6 | kroB200 | 2 | 16127 | 92.48 | 18758 | 47.00 | 16.3 |
| | 3 | 2663 | 42.67 | 2870 | 22.78 | 7.7 | | 3 | 11814 | 103.17 | 13199 | 46.13 | 11.7 |
| | 5 | 1915 | 75.00 | 1994 | 22.66 | 4.1 | | 5 | 8525 | 126.04 | 10618 | 46.83 | 24.6 |
| | 7 | 1661 | 66.54 | 1603 | 23.10 | -3.6 | | 7 | 7720 | 120.19 | 8678 | 46.41 | 12.4 |
| kroA150 | 2 | 15326 | 40.34 | 16578 | 22.77 | 8.2 | ts225 | 2 | 70660 | 230.60 | 71594 | 64.01 | 1.3 |
| | 3 | 10810 | 47.01 | 12600 | 22.83 | 16.6 | | 3 | 55237 | 245.30 | 52330 | 63.12 | -5.6 |
| | 5 | 7323 | 54.11 | 8567 | 23.05 | 17.0 | | 5 | 38748 | 189.31 | 31318 | 64.13 | -23.7 |
| | 7 | 6290 | 42.09 | 7901 | 22.66 | 25.6 | | 7 | 35460 | 213.09 | 27921 | 63.53 | -27.0 |
| kroB150 | 2 | 14633 | 54.68 | 18342 | 22.84 | 25.3 | tsp225 | 2 | 2145 | 277.23 | 2481 | 62.39 | 15.7 |
| | 3 | 10565 | 36.80 | 11512 | 22.91 | 8.9 | | 3 | 1612 | 145.43 | 1743 | 63.56 | 8.1 |
| | 5 | 7257 | 35.87 | 7539 | 22.82 | 3.9 | | 5 | 1183 | 184.62 | 1308 | 63.83 | 10.6 |
| | 7 | 8221 | 31.13 | 6057 | 22.92 | -35.7 | | 7 | 1044 | 209.54 | 1016 | 63.89 | -2.7 |
| pr152 | 2 | 43922 | 52.08 | 51982 | 23.71 | 18.3 | | | | | | | |
| | 3 | 38752 | 50.79 | 48238 | 23.35 | 24.5 | | | | | | | |
| | 5 | 33006 | 91.95 | 43940 | 23.27 | 33.1 | | | | | | | |
| | 7 | 32245 | 102.58 | 41624 | 23.57 | 29.1 | | | | | | | |

### A.3.3 CVRP

Table 10 shows results from CVRPLib (Uchoa et al., 2017). Those values are the total traveling distance. Although our model is trained to minimize makespan (MinMax), our model shows better performance than AM in the total distance (MinSum).

Table 10: Performance comparison on CVRP library

| Instance | Opt | OR-Tools | AM (N=1280) | AM (N=5000) | Ours |
|---|---|---|---|---|---|
| X-n101-k25 | 27591 | 29405 | 39437 | 37702 | 39133 |
| X-n106-k14 | 26362 | 27343 | 28320 | 28473 | 27940 |
| X-n110-k13 | 14971 | 16149 | 15627 | 15443 | 17388 |
| X-n115-k10 | 12747 | 13320 | 13917 | 13745 | 19292 |
| X-n120-k6 | 13332 | 14242 | 14056 | 13937 | 16589 |
| X-n125-k30 | 55539 | 58665 | 75681 | 75067 | 69919 |
| X-n129-k18 | 28940 | 31361 | 30399 | 30176 | 36649 |
| X-n134-k13 | 10916 | 13275 | 13795 | 13619 | 14800 |
| X-n139-k10 | 13590 | 15223 | 14293 | 14215 | 16368 |
| X-n143-k7 | 15700 | 17470 | 17414 | 17397 | 23548 |
| X-n148-k46 | 43448 | 46836 | 79611 | 79514 | 62240 |
| X-n153-k22 | 21220 | 22919 | 38423 | 37938 | 33079 |
| X-n157-k13 | 16876 | 17309 | 21702 | 21330 | 19702 |
| X-n162-k11 | 14138 | 15030 | 15108 | 15085 | 19491 |
| X-n167-k10 | 20557 | 22477 | 22365 | 22285 | 25676 |
| X-n172-k51 | 45607 | 50505 | 86186 | 87809 | 63191 |
| X-n176-k26 | 47812 | 52111 | 58107 | 58178 | 66997 |
| X-n181-k23 | 25569 | 26321 | 27828 | 27520 | 27220 |
| X-n186-k15 | 24145 | 26017 | 25917 | 25757 | 30086 |
| X-n190-k8 | 16980 | 18088 | 37820 | 36383 | 18651 |
| X-n195-k51 | 44225 | 50311 | 79594 | 79276 | 66957 |
| X-n200-k36 | 58578 | 61009 | 78679 | 76477 | 68502 |
| Gap | - | 8.06 | 32.97 | 31.62 | 30.22 |

## A.4 GRAPHICAL SOLUTIONS EXAMPLE

This section provides the visual results for several vehicle routing problems solved by the trained policy.

### A.4.1 TRAINING RESULTS ON MCVRP

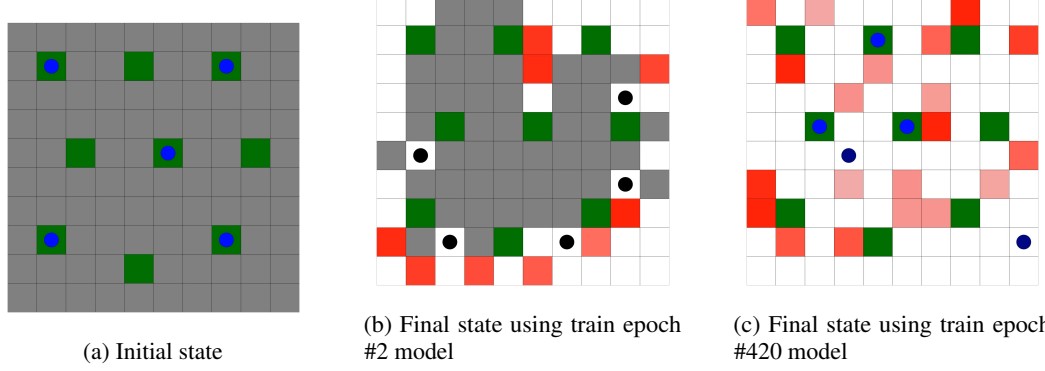

(a) Initial state

(b) Final state using train epoch #2 model

(c) Final state using train epoch #420 model

Figure 3: $N_c = 100, N_v = 5$ case of mCVRP

Figure 3 depicts the evaluation case ($N_c = 100, N_v = 5$ case of mCVRP) during training process. This test environment is designed to mimic the search and rescue problems, operating multiple drones to search victims distributed over a map.

Green grids are for refueling nodes, blue circles are for vehicles (color becomes black as fuel capacity diminishes), and grey grids are for unvisited customers. If a customer is visited by a vehicle, the color becomes red or white; the red girds indicate the victims, while the white indicates no victim in that cell. These red grids do not have any effect on performance measure, because visiting all customers (grey grids) is top-priority.

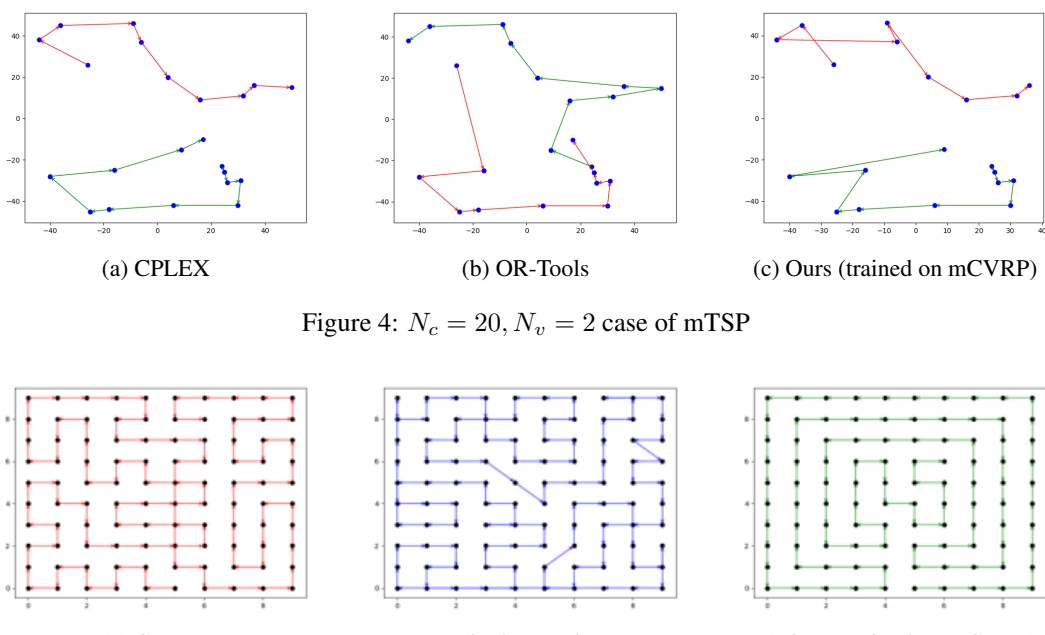

| (a) CPLEX | (b) OR-Tools | (c) Ours (trained on mCVRP) |

Figure 4: $N_c = 20, N_v = 2$ case of mTSP

| (a) CPLEX | (b) OR-Tools | (c) Ours (trained on mCVRP) |

Figure 5: TSP with a single vehicle

At the very beginning of the training process (figure 3 (b)), the trained model cannot have vehicles search over the cells and make them die due to lack of fuel. At the end of the training process (figure 3 (c)), however, all customers are visited by the vehicles.

### A.4.2 MVRP (MTSP)

Figure 4 depicts an example solutions of mTSP using CPLEX, OR-Tools, and RL. The RL-based solver is trained under mCVRP environment but we solve mTSP with the trained model.

### A.4.3 TSP

Figure 5 depicts an example solutions of TSP using CPLEX, OR-Tools, and RL. The RL-based solver is trained under mCVRP environment but we solve TSP with the trained model.

### A.5 NETWORK ARCHITECTURE AND HYPERPARAMETERS

Table 11 is summary of hyperparameters used in training process.

Table 11: Hyperparameters

| Parameters (for PPO update) | Value | Parameters (for GRLTS) | Value |
|---|---|---|---|
| Optimizer | Adam | MLP units | $(32, 32)$ |
| betas $(\beta_1, \beta_2)$ | $(0.9, 0.999)$ | Neuron initialization | Kaiming normal |
| Learning rate $(\eta_{policy})$ | $2 \times 10^{-4}$ | Activation fn. | ReLU |
| Learning rate $(\eta_{Q-network})$ | $2 \times 10^{-4}$ | Node umbedding epoch $(N^{hop})$ | 5 |
| gamma $(\gamma)$ | 0.99 | Node feature dim. | 5 |
| clip ratio $(\epsilon)$ | 0.2 | Edge feature dim. | 10 |
| value function coeffi. | 0.5 | Max. decision epoch | $3 \times N_c$ |
| entropy coeffi. | 0.01 | | |
| PPO update epoch $(K)$ | 10 | | |
| Training epoch | 1 | | |

## A.6 TRAINING PERFORMANCE

Figure (6) shows how the trained policy perform on the thee set of validation problems while the policy is being trained by the random mCVRP inc stances. The three validation problems are 1) 100 customers covered by 5 vehicles, 2) 100 customers by 10 vehicles and 3) 400 customers covered by 20 vehicles. The first row show the performance on the random training instances while the second, third, and fourth row show the performances on the three validation problems.

As training processes progress, the trained model gradually becomes more efficient, that is, a fleet of vehicles visits more customers faster. After about 200 training epochs, the trained model converges, but there are some randomness due to the nature of policy gradient structure. Although the makespan curves in the second column of training case seems relatively constant (the first row row), the level of difficulty is increasing as curriculum becomes harder (The number of refueling is also increasing as curriculum becomes harder).

Comparison of case #1 and case #2 give us two interpretations: 1) case #1 shows the model is trained to visit all customers and 2) in case #2, the model achieves consistent performance as training iterations progress in spite of same visit ratio as 1 around 100 epoch and 350 epoch. Besides, the makespans of both training epochs are very close. Case #2 is relatively easier environment than case #1 in that more vehicles are deployed to serve same number of customers.

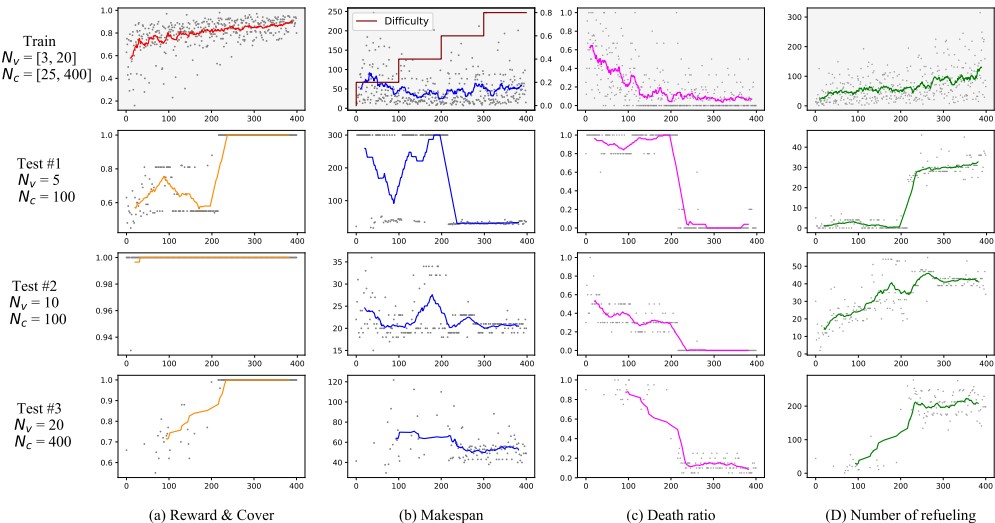

Figure 6: Performance curves of training (1st row) and testing (2nd, 3rd and 4th rows). Note that we plot a reward curve and cover ratio curves in 1st column for training and tesing, respectively.

## A.7 PSEUDO CODE

**Simulation: Problem Instance generation** We generate random mCVRP instances and use these training instances to train the proposed model. For training, we randomly select the number of vehicles, customers, and refueling stations. We randomly assign these entities over the grid world with the distance between each grid cell being 1. When transforming the snapshot of the mCVRP to a graph, we use the Manhattan distance between the two cells as the edge weight. In addition, the speed of each vehicle is set to be 1; thus, each vehicle moves over one cell during a single time-tic. Although the world is represented as discrete gird, the mCVRP problem's state transition is event-based. Whenever the agent reaches the assigned customer, the event occurs, and all the edge distance is updated.

Initially, this grid world is developed to use the trained policy for search and rescue problems, seeking distributed victims over the grid-world. Because each vehicle (drone) can search a specific zone within a certain amount of time, using a cell-based grid world is a reasonable choice.

Once the policy is trained, it can be used for both a discrete world or a continuous world because these environments can be transformed into a graph without any difference. We validated the proposed method using CVRP and TSP instances defined over the continuous state to compare the performance of the proposed model to other deep baseline line models. Other deep baseline line models all use the continuous state.

To boost training, we vary the difficulty level of random mCVRP instances during training process (see Algorithm (2)). For the curriculum instance generation in line 9, we compute $N_{curr}$, the number of customers which are assumed to be *visited* already, as $N_{curr} = N_c \times (1 - 2 \times currLevel/10)$. Then, we choose $N_{curr}$ tasks randomly and mark as $visited$ and distribute $N_{curr}$ for all agents' visit number $q^t$.

---

**Algorithm 2** Training instance generation

---

1: Generate random problem instances $N_v$, $N_c$, $N_r$, $x_v^0$, $x_r^0$,
2: $currLevel \leftarrow 0$
3: **for** $i_{training} = 1, \ldots, N_{training}$ **do**
4:     **for** each 100 training iterations **do**
5:         $currLevel \leftarrow currLevel + 1$
6:     **end for**
7:     Generate a random number $p_{curr} \sim U[0, 1]$
8:     **if** $p_{curr} \geq 0.5$ **then**
9:         Curriculum instance generation
10:     **else**
11:         Non-curriculum instance generation (followed by line 1)
12:     **end if**
13: **end for**

---

**Simulation: Episode generation** As a medium for the interaction of the two components such as environment and the GRLTS, simulation conducts the agents' actions and stores the state transitions which are used to update the GRLTS. Starting from the problem instance generation (Algorithm (2)), simulation assigns the agents' assignment as $a_{agt}^t$ using the action probability $\pi(\cdot|o_{agt}^t)$ computed from the proposed GRLTS. Simulation computes the agent $agt$'s transition time $\Delta_{agt}$ as follow:

$$\Delta_{agt} = {}^{Dist(x_{agt}^t, a_{agt}^t)}\!/_{v_{agt}} = Dist(x_{agt}^t, a_{agt}^t) \tag{13}$$

where velocity of agents are constant as 1 and distance between $i$ and $j$ $Dist(i, j)$ can be the Manhattan distance or Euclidean distance.

---

**Algorithm 3** Episode generation

---

1: Initialize problem instances (based on Algorithm 2)
2: $termination, done \leftarrow False, t \leftarrow 0, memory \leftarrow \emptyset$
3: $idleAgts \leftarrow \{1, 2, \ldots, N_v\}, actingAgts \leftarrow \emptyset$
4: **while** $termination$ **do**
5:     **for** $agt \in idleAgts$ **do**
6:         Send an obervation $\Pi_{agt} o_{agt}^t$ to a GRLTS
7:         Choose an action $a_{agt}^t \sim \pi(\cdot | o_{agt}^t)$ where an action probability is from GRLTS
8:         Assign the action $a_{agt}^t$ to the agent
9:     **end for**
10:    Compute and update transition time $\Delta_{agt}$ for $agt \in idleAgts$ (Equation (13))
11:    $actingAgts \leftarrow \{agt | \Delta_{agt} = \min_{agt} \Delta_{agt}\}$ for all agents
12:    $t \leftarrow t + \min_{agt} \Delta_{agt}$
13:    **for** $agt \in actingAgts$ **do**
14:       Perform the recently assigned action $a_{agt}^{t'}$ assigned by $\pi(\cdot | o_{agt}^{t'})$
15:       Observe the observation $o_{agt}^t$, reward $r_{agt}^t$ and $done$
16:       Append a state transition tuple $< t', o_{agt}^{t'}, a_{agt}^{t'}, r_{agt}^t, o_{agt}^t >$ to $memory$
17:    **end for**
18:    **if** $done$ **then**
19:       $termination \leftarrow True$
20:    **else**
21:       $idleAgts \leftarrow actingAgts$
22:    **end if**
23: **end while**

---

**RL networks: GRLTS**     Most of algorithms in the RL model is explained the paper so that we focus on the interaction between simulation and GRLTS networks and internal message exchanges.

---

**Algorithm 4** GRLTS

---

1: Receive an observation $\Pi_{agt} o_{agt}^t$ at time $t$ from simulation (line 6 of Algorithm 3)
2: Generate $\mathcal{G}^{t,(0)}$ with an input of observations $\Pi_i o_i^t$
3: **for** node embedding iteration $\tau = 1, \ldots, N_{hop}$ **do**
4:    Compute connectivity $C_{ij}$ for all nodes $i, j$:

    For agent nodes $n$, $C_{nj} = \begin{cases} f^n, & \forall j \in V_T \cup V_R \\ \infty, & \forall j \in V_A. \end{cases}$

    For refueling nodes $r$, $C_{rj} = F, \forall j \in V$
    For task nodes $t$, $C_{tj} = 5, \forall j \in V$
5:    Store the node embedding vector $h_i^A$ at $memeory_i$
6:    Update the connected edges following Equation (3)
7:    Aggregate the incoming edge featured at each nodes following Equation (4)
8:    Update the node feature with the node updator function $\phi_v$
9: **end for**
10: Compute $Q(h_i^A, a)$ for agent $i$ where $a \in A(o_i) = \{v \in \{V_R \cup V_T\} | C_{iv} = 1\}$
11: Compute $\pi(a|o_i) = \frac{\exp(Q(h_i^A, a))}{\sum_{a \in A(o_i)} \exp(Q(h_i^A, a))}$
12: Send the action probability $\pi(a|o_i)$ to simulation

---

