# OpenReview forum: "Learning a Transferable Scheduling Policy for Various Vehicle Routing Problems based on Graph-centric Representation Learning"
_ICLR.cc/2021/Conference — Reject_

### Official Review · AnonReviewer3 · 2020-10-28
**RL approach to solve various VRPs**

**Rating:** 5
**Confidence:** 5

**Review:**

Summary
--------------
The paper presents a reinforcement learning approach to learn a routing policy for a family of Vehicle Routing Problems (VRPs). More precisely, the authors train a model for the min-max capacitated multi vehicle routing problem (mCVRP), then use it to solve variants of the problem that correspond to various VRP problems (with a single vehicle, no capacity constraints, no fueling stations, etc). They use a GNN to represent the states and the PPO algorithm to learn the policy. They validate their approach on both random instances and literature benchmarks.

Strong points
-------------------
1. The goal of using a single policy to solve a variety of routing problems
2. First RL-based approach to tackle the multiple vehicle setting of VRPs
3. Extensive numerical experiments on both randomly generated and benchmark instances


Weak points
-----------------
4. There are a lot of imprecisions/typos/lack of definitions/mathematical imprecisions (see Feedback to improve the paper)
5. According to the definition of the rewards, the expected return of the policy would be: $\sum_v \sum_t (r^v_{visit} + r^v_{refuel})$. It is important to note that this does not correspond to the objective function of the mCRVP problem. This choice of reward does not look natural to me and it would be useful to better motivate it.
6. Sec 3.3: “When an vehicle node i reaches the assigned customer node, an event occurs and the vehicle node i computes its node embedding and selects one of its feasible action…”. This is crucial but not clear to me. At a step t, some vehicles might still be in between two cities. How is that taken into account in the state s_t? With the definition of the transition function, I understand that when an action is taken at t, the vehicle arrives at destination at t+1. Does it mean that you sequentially assign only one vehicle to a city and then “wait for it to arrive” before computing the next assignment? In that case what are the events about?
7. Tables 1, 2, 3: to be relevant the results should averages over a number of random instances. Maybe it’s already the case but it is not mentioned.
8. Sec 3.4 about the training should be more precise. It was difficult for me to find the relevant information because it was scattered at different places in the (10 pages) appendix.
9. The authors do not mention whether they will share their code.

Recommendation
-------------------------
I would vote for reject. Although the problem addressed is interesting, the paper is not well-written and there are too many typos and missing explanations to understand the method.

Arguments for recommendation:
----------------------------------------------
see weak points


Questions to authors
-----------------------------
10. What prevents a vehicle from getting to a customer and then not having enough fuel to go back to a refuelling station?
11. Table 1: are the results averaged over a number of random instances?
12. To improve the results on TSP and VRP, have you tried including instances with only 1 vehicle during training?



Feedback to help improve the paper
---------------------------------------------------
13. In the formal definition of the mCRVP (Section 2.1), there is no mention that the customers should all be visited. As it is presented, the problem is solved trivially by all vehicles doing nothing.
14. In the state definition, in the vehicle state “x_v^t is the allocated node that vehicle v to sist”. This is important and not clear. I think the authors mean that it’s the node where the vehicle v is currently located. But then in customer state, x_c^t is a location. V^c should depend on t.
15. Action a_t in {V_c union V_R} is not mathematically correct with the definition of V_C and V_R. The update of the v_t^c variable is missing (although in definition there was no dependence on t)
16. In the definition of the refuelling rewards (section 2.2.3)
(a) the index t is missing in q^v. The rewards should also be dependent on t.
(b) What values for \alpha? If F_v < 1 then the reward is negative or might be undefined if F_v =1
17. Section 2.3: TSP is a special case of mCRVP. The fact that there is no fuel constraint is missing
18. Equation (2) what is R_C. How is it chosen?
19. Equation (3) R_R = F^v for which v?
20. Right after equation (5),\alpha_ij is defined as softmax(e_ij) but then e_ij is defined as a function of \alpha
21. Equation 7, I believe the sum should be over N_C(i) \union N_R(i)

---

> ### Author Response · Authors · 2020-11-24
> **Response to "Feedback to help improve the paper 2"**
>
> [18  Equation (2) what is R_C. How is it chosen?]
>
> $R_C$ is the radius defining the scope of a customer node $c\in V_c$. We can interpret $R_C$ as the spatial range where customer nodes influence each other. Each customer node processes its own state information and sends the processed message to other customer nodes located within $R_C$ and, at the same time, receives the messages from the customer nodes located within $R_C$. The customer node then processes its own state information and the received messages to compute the updated node embedding. This computed node embedding can represent the current node's potential, summarizing the relative geographic information computed in terms of the current customer node. The node embedding for the customer nodes is then used by vehicles to select the next customer node to visit.
> $R_C$ is one of hyperparameter balancing between the representability of graph state and computational cost; larger $R_C$ induces more edges and, thus, many massage passing during computing node embedding.
> We set  $R_C=10$ while following a typical hyperparameter selection procedure. We select $R_C=5$ such that it achieves the highest accumulated reward on the validation test problem instances. In the revised manuscript, we clearly mention this fact.
>
>
> [19  Equation (3) R_R = F^v for which v?]
>
> In this study, we assume that all vehicles are homogeneous. Thus, $F^v=10$ for all vehicles. Owing to your feedback, we realized that using $R_R = F^v$ is not a good choice. In the revised manuscript, we set $R_R = \max_{v \in V_V} F^v$ to ensure $R_R$ is uniquely determined.
>
> [20 Right after equation (5),\alpha_ij is defined as softmax(e_ij) but then e_ij is defined as a function of \alpha]
>
> $e_{ij}=f_\alpha(s_i, s_j;w_\alpha)$, which is used for computing the attention coefficient $\alpha_{ij}$, is computed using function $f_\alpha$. This function $f_\alpha$ is not a function of $\alpha$. It is the neural network (MLP) parameterized by $w_\alpha$ and accepts as inputs the state information, $s_i$ and $s_j$, for the two nodes and compute $e_{ij}$.
> To prevent such confusion, we replace $e_{ij}=f_\alpha(s_i, s_j;w_\alpha)$ to $e_{ij}=f_e(s_i, s_j; W_e)$ in the revised manuscript.
>
> [21 Equation 7, I believe the sum should be over N_C(i) \union N_R(i)]
>
> That is true. Thank you for the correction. During coding implementation, we actually summed over $\mathcal{N}_V(i) \cup \mathcal{N}_R(i)$ to reduce the computation. We have modified Equation 7 in the revised manuscript.

---

> ### Author Response · Authors · 2020-11-24
> **Response to "Feedback to help improve the paper 1"**
>
> [13  In the formal definition of the mCRVP (Section 2.1), there is no mention that the customers should all be visited. As it is presented, the problem is solved trivially by all vehicles doing nothing.]
>
> In the mathematical formulation for mCVRP problem, the optimization variables that we have to determine are defined as $x_{a,i,j}\in{0,1} \forall a \in A, \forall i \in V, \forall j \in V$.  $x_{a,i,j}=1$ indicates that agent $a$ travels from node $i$ to node $j$. In mCVRP formulation, we has the constraint $\sum_{j \in V} x_{aij} = 1, \forall a \in A, i \in vStart_a : i \neq j$ requiring every customer node should be visited exactly once by a single vehicle.
> The proposed approach derives the data-driven decision making policy that sequentially assigns an idle vehicle to one of the unvisited customer nodes by applying RL. Because the policy is solely trained by using a reward signal, we just design an effective reward signal (incentive) to induce vehicles to visit unvisited customers as many as possible. In RL, we cannot define the status “problem solved.” We just carefully formulated MDP for mCVRP such that the maximum accumulated reward achieved by all agents coincides with the “solved status” of mCVP (all customers were visited). Please note that since the designed reward signal provides an incentive for a vehicle to visit one of the unvisited customer nodes, vehicles will eventually visit all the customers to solve the target mCVRP problem.
>
> [14  In the state definition, in the vehicle state “x_v^t is the allocated node that vehicle v to visit”. This is important and not clear. I think the authors mean that it’s the node where the vehicle v is currently located. But then in customer state, x_c^t is a location. V^c should depend on t.]
>
> The state of vehicle is defined as $s_t^v=(x_t^v, f_t^v, q_t^v)$. When a vehicle reaches a destination node (customer), all other vehicles are under traveling toward their assigned customers. For such a case, $x_t^v$ denotes the target customer node that the vehicle $v$ is currently heading; in other words, it can be considered as the scheduled (reserved) destination of vehicle $v$.
> On the other hand, in the customer’s state $s^c_t = (x^c, v^c)$, $x^c$ is the location of customer node $c$ which does not change with time. A customer is located at the fixed location $x^c$ to be served by a vehicle (it does not move with a vehicle like a taxi-rider). You can imagine that a customer in this study is like a customer who is waiting for a parcel delivery service.
>
> [15  Action a_t in {V_c union V_R} is not mathematically correct with the definition of V_C and V_R. The update of the v_t^c variable is missing (although in definition there was no dependence on t)]
>
> In the state $s^c_t = (x^c, v^c)$ of the customer node $c$, the location $x^c$ of customer $c$ does not change with the time as explained above. All customers are located at fixed points, waiting for a vehicle to serve them. In that sense, the action $a_t^v$ of vehicle $v$ is choosing either one of unvisited (unserved) customer nodes or refueling stations.
>
> [16  In the definition of the refueling rewards (section 2.2.3) (a) the index t is missing in q^v. The rewards should also be dependent on t. (b) What values for \alpha? If F_v < 1 then the reward is negative or might be undefined if F_v =1]
>
> Thank you for correcting the mistake. The refueling reward is has been corrected as $r^{v}_{refuel} = q^{v}_t \times (\frac{(F^{v} - f^{v}_t)}{(F^{v}-1)})^{\alpha}$.
> The fuel capacity $F^v$ of vehicle $v$ is defined as 10, which is the equivalent to the total traveling distance that vehicle $v$ can travel with the fuel tank fully loaded (We added this information to the revised manuscript). In addition, we set $\alpha=2$.
>
> [17 Section 2.3: TSP is a special case of mCRVP. The fact that there is no fuel constraint is missing.]
>
> In the mixed-integer linear programming (MILP) formulation for mTSP and TSP (stated in Appendix), we clearly dropped the fuel constraint. However, in the main text, the relaxation of the fuel constraint is not clearly described. We modified the description of mTSP and TSP in the main text as follows:
> “TSP is the problem where a single vehicle is operated to serve every customer while minimizing the total traveling distance. The agent needs or needs not come back to the depot. This problem does not have capacity constraints.”

---

> ### Author Response · Authors · 2020-11-24
> **Response to "Question to authors"**
>
> [What prevents a vehicle from getting to a customer and then not having enough fuel to go back to a refueling station?]
>
> We induce a strategic refueling strategy using RL training with carefully designed reward signals: First, we use customer visit reward $r^{v}_{visit} = q^{v}_t$. This reward is provided when an agent visits a customer; the more customer nodes a vehicle $v$ visits, the greater reward it can earn due to the incrementally increasing visit count $q^{v}_t$. For example, on the first visit, a vehicle receives only one reward score. However, on the 20th visit, the vehicle receives a 20 reward score. Thus, to maximize the accumulated reward over long-term horizons, vehicles should learn to refuel from time to time. Here, to help vehicle’s decision making regarding refueling, we provide a vehicle as state information the current fuel level, distances to neighboring unvisited customer nodes, and distances to neighboring refueling stations in a graph form. By configuring the relative distances to these nodes and their current fuel level, a vehicle can learn how to make strategic decisions regarding refueling.
>
> Second, we additionally use refueling reward $r^{v}_{refuel} = q^{v}_t \times (\frac{(F^{v} - f^{v}_t)}{(F^{v}-1)})^{\alpha}$ to give vehicles an incentive bonus proportional to the amount of fuel they refill.
>
> It is possible for a vehicle can get a certain amount of reward by just roaming around refueling stations; however, the reward collected by such repetitive refueling is significantly lower than the accumulated reward collected by visiting a series of customer nodes. Thus, vehicles will eventually learn to strategically refuel to maximize the accumulated total reward in the long run, which we have proven while conducting a series of experiments.
>
>
>
> [Table 1: are the results averaged over a number of random instances?]
>
> The previous results were collected by solving a single instance of mCVRP. Because it takes a significant amount of time to compute the optimum solution for an mCVRP instance using CPLEX, we just solve one mCVRP instance using CPLEX, Google ORT, and our proposed method. Since there is no exact solution nor well-established baseline algorithms for mCVRP (and mTSP), we decided to rely on the exact solution of CPLEX. However, we realized that this strategy induces another issue of statistically generalizing the result. On the other hand, the results in Table 4 and Table 5 were computed using 100 random test CVRP and TSP instances. We do not have to use CPLEX to compute the exact solution for these problems but can use the well-known algorithms that can quickly compute the near-optimum solutions as baseline algorithms.
>
> Since providing statistically sound results for all target problems coherently is essential, we decided to produce results for mCVRP and mTSP in a similar manner with CVRP and TSP. That is, we solve 100 test mCVRP and mTSP instances, which are randomly generated, and provide the average performance and standard error for that. Because the makespan itself varies for different problem instances, we use as a performance metric the relative ratio between the makespan computed by the proposed method and the makespan calculated by Google ORT. That is the number in Table 1, 2, and 3 are now the average performance ratio indicating how much the proposed method is doing well compared to Google ORT. Please note that we do not use CPLEX solution as a baseline anymore in these tables. We still provide results obtained by CPLEX in the Appendix.
>
> [To improve the results on TSP and VRP, have you tried including instances with only 1 vehicle during training?]
>
> During training the policy for mCVRP, we are already using the random mCVRP instances having $m=1$ while considering combinations of random numbers $m$ ranging from 1 to 10 and $n$ ranging from 10 to 100. In the revised manuscript, we now clearly describe the generation procedure for training instances.

---

> ### Author Response · Authors · 2020-11-24
> **Response to "Weak Points"**
>
> [4. There are a lot of imprecisions/typos/lack of definitions/mathematical imprecisions (see Feedback to improve the paper)]
>
> We have carefully revised to correct these issues. According to your feedback, we have provided the precise definition of the mathematical notations and terms in the revised and updated manuscript.
>
> [5. Question on the definition of the reward]
>
> The primary goal of the various routing problem is to minimize the total completion time. For single-vehicle routing problems (TSP and CVRP), minimizing the total completion time can be translated into minimizing the sum of total accumulated traveling distance (because the traveling distance is proportional to the total completion time when the speed is a constant). However, for multiple vehicle routing problems (mTSP and mCVRP), minimizing the total completion time cannot be achieved by merely minimizing the sum of the traveling distance of vehicles.
> The straight and the most direct reward function would be the actual total completion time by multiple vehicles. However, such a reward signal is very difficult to be used for inducing the cooperative and strategic behaviors of multiple vehicles in a decentralized manner due to the following reasons:
>
> •	This reward can only be obtained only after a single episode finishes (delayed reward), making the learning very slow.
>
> •	It is challenging to redistribute this scalar reward, which is delayed and sparse, over the entire time horizon and over multiple agents. In other words, the algorithm needs to reassign the episodic reward over time and over agents to correctly evaluate the goodness of a particular agent’s action at a specific time in terms of minimizing the makespan would be extremely difficult.
>
>
> To resolve these issues, we carefully design an immediate reward signal that can be given to every agent at every time such that maximizing the accumulated reward signal over agents and time indeed induces the minimized total completion time. To this end, we use the following two immediate rewards:
>
> •	The visiting reward signal is our choice to achieve such an objective. Because when an agent visits (serves) more customers, it receives a higher reward proportional to its visit count so far. If the agent visits a city first, it will receive 1; however, if the agent visits the same city after vising the other nine cities (10th visit), the agent receives an immediate reward of 10. This incentivized visiting reward induces all agents to visit as many cities as possible and as quickly as possible. In short, we induce multiple agents' cooperative behavior by making individual agents pursue their own interests.
>
> •	The refueling reward is to ensure that each agent satisfies fuel constraints.
>
>
>
> [6. Does it mean that you sequentially assign only one vehicle to a city and then “wait for it to arrive” before computing the next assignment? In that case what are the events about?]
>
> When agent $i$ reaches the scheduled customer node, other vehicles are under moving toward their target customers. When agent $i$ selects the next customer node, it should not select the customer nodes that are supposed to be served by other vehicles approaching them. Such reserved actions by other agents are represented by vehicle state information $s_t^v=(x_t^v, f_t^v, q_t^v)$; $x_t^v$ is the allocated node that vehicle $v$ to visit and considered to define the feasible actions $A_i(s_t)$ for agent $i$.
>
> In the formulation, an event is defined as the time when a vehicle reaches a customer (or a refueling station) and becomes idle. For every event, the proposed algorithm assigns a single idle vehicle to another feasible customer or refueling station. Thus, we use an event-based transition for the MDP formulation of mCVRP. One restriction we impose for this formulation is that once a vehicle is assigned to a customer, it cannot be changed.
>
> [7. Tables 1, 2, 3: to be relevant the results should average over a number of random instances. Maybe it’s already the case but it is not mentioned]
> The response to these comments is provided by answering to your question 11. Now, all the results are averages over 100 random instances.
>
> [8. Sec 3.4 about the training should be more precise. It was difficult for me to find the relevant information because it was scattered at different places in the (10 pages) appendix]
>
> We have revised the main text to include the detailed training procedure. The problem formulation for learning and the associated algorithm to derive the policy is now discussed in the main text. We kindly request to check the newly uploaded manuscript.
>
> [9. The authors do not mention whether they will share their code]
> We will open up a GitHub repository archiving all the codes and simulation environments. In addition, by providing the benchmark problems and the associated optimization results, we will allow researchers to solve the same problem with more advanced algorithms and compare the performance with ours.

---

### Official Review · AnonReviewer1 · 2020-11-01
**An RL algorithm for mCVRP problem is proposed**

**Rating:** 6
**Confidence:** 5

**Review:**


An end-to-end RL algorithm, Graph-centric RL-based Transferable Scheduler (GRLTS) is proposed to solve the capacitated multi-vehicle VRP problem.
The fuel capacity of vehicles is considered and vehicles can visit charging stations in the middle of their routes toward the customers, and if their charge is about to finish, they get a reward as good as visiting a customer.
The goal is to minimize the makespan (completion time), which is the time between starting and finishing visiting all customers.
The state of each vehicle includes the current fuel level, the number of customers served up to now, the allocated node to visit. The state of customers involves its location and a visit indicator flag. And, the state of each charging station is its location. The action for each vehicle is the next node to visit, and the reward consists of two parts: (i) the number of visited customers, (ii) a reward to refuel the vehicle worth visiting a customer node when the battery is almost empty, otherwise a small value.

To use the proposed states, node and edge embedding are done by following GNN, a graph attention network that learns the attention weight over the neighbor nodes/edges to build the representation of each node. The edge values are considered a message that the source node sends to the target node. Also, in the GNN only the neighbor nodes/edges are considered.


Major comment:
Q1- By just reading the main body of the paper, it is not clear what RL model you have used, and what does it mean to obtain a softmax operator over the Q-values in equation (7). If it is a regular actor-critic model, you do not need the critic for decision making, and the critic is only involved in the training. I am not sure why the critic is involved in decision making in equation (7). Can you please clarify?

Q2- GRLTS does not perform better than the benchmarks in CVRP and TSP. What is the point of having those results in the main body of the paper?
To me, the current paper does not give any idea about your training algorithm and just provides the MDP definition and the result. I would suggest moving them to the appendix and explain some details of your training algorithms in the main body instead of the appendix.

Q3- Why Table 6 and Table 7 does not involve other RL and heuristic algorithms? It looks like a cherry-picking among the benchmark algorithms is performs.

---

> ### Author Response · Authors · 2020-11-24
> **Response to Q3**
>
> For single-agent vehicle routing problems (TSP, CVRP), several RL approaches have been employed to solve the target benchmark problems. For such cases, we have included all the RL baseline models in Table 6 and Table 8. However, no RL-based approach has been suggested for multi-vehicle routing problems. Thus, we did not include any RL approach's performance in table 7 for the mTSP benchmark problem. Please note that there are no benchmark problems for mCVRP because we have newly formulated this routing problem considering multiple vehicles and their constraints.
>
> •	TSP (Table 6): Several RL approaches have been used to solve the same benchmark problems. Thus, we have included these RL approaches (Drori et al., (2020) and GPN, S2V-DQN).
> •	mTSP (table 7): no RL methods have been employed previously to solve such benchmark problems
> •	CVRP (table 8): Several RL approaches are used to solve the same benchmark problems. Thus, we have included these RL approaches (AM).

---

> ### Author Response · Authors · 2020-11-24
> **Response to Q2**
>
> We have expanded the description of the learning algorithm in the main text. We kindly request the reviewers to check the revised manuscript.
>
> Because showing the trained model can solve reasonably well CVRP and TSP instances without re-training is the primary contribution of the current study, we leave the results in the main text.

---

> ### Author Response · Authors · 2020-11-24
> **Response to Q1**
>
> The term critic $Q(h_i^H, h_k^H;\phi)$  is misleading; it is different from the critic function used in the actor-critic framework. We should not use the term of the critic. We replace $Q(h_i^H, h_k^H;\phi)$ into $F(h_i^H, h_k^H;\phi)$ representing the fitness function quantifying the goodness for agent $i$ to choose the particular destination $k$.
>
> Utilizing this fitness function, we structured our policy network such that it can select one of the best action among the feasible alternatives. For example, when vehicle $i$ select its next destination, it evaluates the goodness of choosing particular destination $k$ using the fitness function $F(h_i^H, h_k^H;\phi)$ for $k\in V_C \cup V)R$. Then, it will choose the destination $j$ with a probability computed by the softmax operator with the computed fitness of the feasible actions.
>
> The structured actor $\pi(a_t^i|s^t;\phi)$ based on the fitness function $F(h_i^H, h_k^H;\phi)$ is then trained by using the PPO algorithm [1] where the critic is additionally trained to train the proposed actor in a stable manner.
>
>
>
>
> [1] John Schulman, Filip Wolski, Prafulla Dhariwal, Alec Radford, and Oleg Klimov. Proximal policy optimization algorithms. arXiv preprint arXiv:1707.06347, 2017

---

### Official Review · AnonReviewer5 · 2020-11-06
**scientifically sound, perhaps not a good fit**

**Rating:** 5
**Confidence:** 3

**Review:**

This paper considers the problem of capacitated vehicle routing which is a famous combinatorial optimization problem that is known to be NP-hard. This paper takes the approach of solving instances of this problem using RL. The goal is this problem is to minimize the maximum time (or makespan objective) for multiple vehicles to complete various tasks subject to fuel constraint. The paper trains a graph embedding from random instances and then show that it solves new instances of this problem with reasonable accuracy. Moreover, they also show that the embedding can be transferred to other related objectives.

The strengths of this paper are as follows.

- Contributions to the literature of RL for combinatorial optimization which has become a recent growing paradigm. In this paradigm, we seek to obtain statistical algorithms (based on RL for instance) to NP-hard problems where the average instance tends to be computationally easy while the hard instances are not abundant. It is important to identify the set of combinatorial optimization problems for which we can surpass computational hardness and this paper adds to this literature.
- Extensive experimentation: The second strength of this paper is that the paper does a good job of extensively evaluating their method on a variety of instances and also other related problems/objectives.

Having said that, the paper has a number of weakness in my opinion.

- I find this paper to be rather incremental since this is not the first paper to study RL algorithms for this problem. The paper does not make a good case for why (yet) another RL based algorithm is useful for this problem and along what dimensions is this algorithm having the most novel improvements? The paper gives two reasons, but I do not find it convincing enough. In particular, it does not really show empirically why prior methods do not extend to this setting and why a new algorithm needs to be proposed.
- I found very little generalizable learnings from this paper that could be useful for the machine learning community. The paper should make a better case for why the results from this paper are important to the ICLR community. It seems like this paper is more suitable for an appropriate operations research journal such as OR/INFORMS. I would like the authors to expand/justify more along these lines. May be the paper brings to light some hard application that could lead to new algorithmic developments?


Overall, I think the paper is scientifically sound. My ratings stem from the fact that it may not be a good fit for ICLR. As I state above, I do not find generalizable results that can appeal to a broad (or even narrow) machine learning audience and will be a better fit for more domain specific venues.

---

> ### Author Response · Authors · 2020-11-24
> **Responses to "generalizable results that can appeal to a broad machine learning community and will be a better fit for more domain specific venues"**
>
> <Appeal to a narrow machine learning audience: RL based solver for combinatorial optimizations>
>
> A wide range of practical manufacturing, transportation, and logistics problems can be cast as combinatorial optimization problems. For example, dispatching vehicles to transports goods or humans can be cast as vehicle routing problems. Also, optimum scheduling of machines in factories to increase productivity and optimum scheduling GPU/CPU operation to train neural networks efficiently can be cast as combinatorial optimization problems.
>
> Combinatorial optimization problems have been core research topics in the operation research community. A vast amount of effort has been devoted to developing efficient solvers (algorithms) to solve such problems. The OR approach focused on understanding the problem structure and utilizing the mathematical properties to find the optimum solution. Due to the combinatorial action space and NP-hardness, the optimum solutions are typically hard to be found in a scalable manner.
>
> AI/ML field has long been tried to resolve such issues by employing learning-based approaches to solve such complicated problems. The RL-based approach has recently been actively employed to develop a solver for various combinatorial optimization problems because the RL approaches do not require the answer to the target problems. Although many researchers are actively exploring this field, numerous hurdles need to be overcome before AI can effectively solve combinatorial problems. These challenges include:
> •	Representing a function over combinatorial action space using deep neural network. Because the small change in the combinatorial input space can result in a significant change in the output. In addition, the approximated function cannot be differentiated with the input, limiting the trained model to be used for decision making.
> •	Exploring over the combinatorial input function is notoriously difficult. Since the continuity assumption is hardly employed, to learn a target function (value function or state-action function) defined over the combinatorial action space is challenging.
>
> The current study raises these challenges that need to be addressed in this field. For AI/ML to be widely used for solving real-world applications that can be cast as general scheduling problems, the community should identify such issues and work together to resolve the issues together.

---

> ### Author Response · Authors · 2020-11-24
> **Response to "Why the results from this paper are important to the ICLR community?"**
>
> <The proposed method is seeking to resolve general and fundamental questions in ML/AI>
>
> Although the target applications look domain-specific, the current study tries to tackles challenging issues that are common and essential for the general ML community, especially for the ICLR community:
> •	Representation learning for solving combinatorial optimization: several papers have empirically proven that RL approaches can successfully solve some combinatorial optimizations. Such success may be due to a neural network's powerful expressivity for representing a complicated target function with a combinatorial input space. However, the community still does not fully understand which neural network structure with certain inductive biases is the most effective for representing a function with a complicated combinatorial input.
> •	Representation learning for multi-agent coordination: Representation learning for modeling relationships among multiple agents has been dramatically improved, especially using graph neural networks. However, the representation learning for modeling the joint constraints among multiple agents is not yet fully explored.
> •	Representation learning for transferability: Flexible input/feature representation can allow a trained model with particular data set can be used for other data sets whose data distribution is different from training data. To develop a universal solver that can solve various types of routing problems (domain transferability), we need to design a flexible representation for modeling essential structural characteristics of different types of routing problems
>
> The novelty of the current study is that we seek to achieve the above three objectives using a well-designed graph representation for general routing problems and an efficient learning strategy based on RL/MARL. That is, the goal of the current study is to develop an efficient representation learning and policy learning for solving various types (transfer learning) of multi-vehicle routing problems (combinatorial optimization) in a decentralized manner (multi-agent reinforcement learning). The current study thus focuses on designing an effective network structure for infusing inductive biases for (1) capturing the structure of combinatorial optimization (combinatorial action space), (2) modeling the interaction among multiple vehicles (multi-agent coordination), and (3) transferring the induced knowledge to different domains (different types of routing problems).

---

> ### Author Response · Authors · 2020-11-24
> **Response to "Why prior methods do not extend to this setting and why a new algorithm needs to be proposed?"**
>
> Several papers have empirically proven that RL approaches can solve some combinatorial optimizations. However, previous RL approaches developed for solving vehicle routing problems have two issues: (1) the derived solver can learn how to route only a single vehicle, and (2) the derived solver can be used to solve only one type of problem. To resolve these issues, we propose a new approach having the following two novelties:
>
> <MARL framework for inducing cooperation among multiple vehicles>
> Scheduling a single vehicle using RL is very straightforward. However, the same approach cannot be directly expanded to solving routing problems for scheduling multiple vehicles. For example, just increasing the dimension of action (for multiple agents) increases the complexities of learning and decision making. Learning the state-action value over high dimensional joint-action would be difficult, and even though it is possible, selecting the joint optimum action over the high dimensional action is typically infeasible.
>
> To resolve these issues, we propose a decentralized decision-making policy for routing multiple vehicles. Shifting from controlling a single vehicle to controlling multiple vehicles requires an entirely different problem-formulation and learning algorithm. We adapt the structure of multi-agent RL to solve the target routing problem in a distributed manner; we assign each vehicle an optimum action utilizing the local state information around the target agent. To this end, we have to train decentralized policies such that the action taken individually by multiple agents (1) satisfies the joint constraint and (2) induces the improved overall performance (minimizing makespan) in a cooperative manner. Therefore, the proposed method employs the MALR framework to solve large-scale combinatorial optimization in a distributed manner. Our approach is the first RL/MARL approach to solve mCVRP in the literature.
>
> <Transferability over different classes of routing problems>
> Second, our approach enables the scheduling policy trained using mCVRP problem instances to solve different types of vehicle routing problems whose problem structures are entirely different (i.e., the constraints and the sizes of vehicles and customers are different). To achieve this transferability, we propose a general graph-representation strategy that can be used for various types of routing problems and an effective reward function that can capture the objectives of such problem classes. Owing to these new developments, the trained policy for mCVRP can be used to solve CVRP, mTSP, and TSP with reasonable accuracy.
>
> Although other studies also prove transferability, their transferability is limited over just different sized problems but within the same class of problem. Our study proposes a more actively transferable strategy that can be used to solve different types of routing problems as well as the different sized problems in the same class.

---

### Decision · Program_Chairs · 2021-01-07
**Final Decision**

**Decision:**

Reject

**Comment:**

This paper looks at a natural application of robust learning for vehicle routing. The paper introduces some new ideas for this RL problem; although the problem has been considered before.  The paper gives a nice algorithm with extensive experimental contributions.

The paper has some shortcomings. The reviewers found there to be a lack of clarity in the mathematical definitions.  Moreover, there were modeling choices that the reviewers felt needed more thorough explanation.   For these reasons, this paper falls below the bar.  The authors are encouraged to revise the manuscript taking these concerns into consideration.